# VisualPrompter: Semantic-Aware Prompt Optimization with Visual Feedback for Text-to-Image Synthesis

**Shiyu Wu**[*,1,2,3]**, Mingzhen Sun**[*,1,3]**, Weining Wang**[1,3]**, Yequan Wang**[†,2,4]**, Jing Liu**[†,1,3]

[1]Institute of Automation, Chinese Academy of Sciences [2]Beijing Academy of Artificial Intelligence [3]University of Chinese Academy of Sciences [4]Peking University
wushiyu2022@ia.ac.cn,tshwangyequan@gmail.com,jliu@nlpr.ia.ac.cn

## ABSTRACT

The notable gap between user-provided and model-preferred prompts poses a significant challenge for generating high-quality images with text-to-image models, compelling the need for prompt engineering. Current studies on prompt engineering can effectively enhance the style and aesthetics of generated images. However, they often neglect the semantic alignment between generated images and user descriptions, resulting in visually appealing but content-wise unsatisfying outputs. In this work, we propose VisualPrompter, a novel training-free prompt engineering framework that refines user inputs to model-preferred sentences. VisualPrompter utilizes an automatic self-reflection module that identifies absent concepts in the generated images, followed by a target-specific prompt optimization mechanism that revises the prompts in a fine-grained manner. By deconstructing prompts, introducing new elements at the atomic semantic level, and then reassembling them, our framework is able to maintain semantic consistency and integrity throughout the optimization process. Extensive experiments demonstrate the effectiveness of VisualPrompter, which achieves new state-of-the-art performance on multiple benchmarks for text-image alignment evaluation. Additionally, our framework features a plug-and-play design, making it highly adaptable to various generative models. Our code is available at https://github.com/teheperinko541/VisualPrompter.

## 1 INTRODUCTION

Text-to-image (T2I) generation task requires generating realistic images that are consistent with textual descriptions. Recently, diffusion-based generative models (Ho et al., 2020; Rombach et al., 2022) have demonstrated remarkable capabilities in generating vivid images from textual descriptions. However, they still face challenges in correctly representing key concepts within user inputs, as shown in Figure 1a. This issue arises from the significant discrepancy between user input prompts and the prompts preferred by the models Hei et al. (2024). To be specific, novice users often provide brief, coarse-grained descriptions, whereas models tend to perform better with detailed, fine-grained prompts, as such data are commonly used during training. Thus, T2I models may struggle to generate satisfactory results when directly fed with user inputs.

Given that a well-crafted prompt can significantly enhance the quality of generated images (Brown et al., 2020), researchers have explored prompt engineering to automatically refine user input prompts (Hao et al., 2023; Rosenman et al., 2024). For instance, Best Prompts (Pavlichenko & Ustalov, 2023) simply appends a few empirically effective keywords to improve image quality. BeautifulPrompt (Cao et al., 2023) finetunes a large language model to serve as a prompt engineer and incorporate reinforcement learning to leverage visual feedback. Despite effectiveness, these methods suffer from three major limitations. First, current studies on T2I prompt engineering (Hao et al., 2023; Rosenman et al., 2024) primarily focus on enhancing the style and aesthetics of gen-

---

[*]Equal Contribution.

[†]Corresponding Authors.

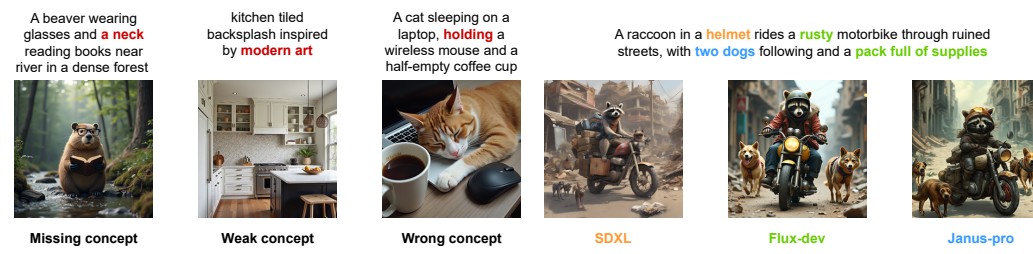

(a) Typical failures of generative models        (b) Same prompt in different incorrect ways

Figure 1: Existing T2I generative models often fail to correctly draw key concepts within the provided prompts. Different models also fail differently on the same prompt.

erated images, neglecting or even compromising the alignment between texts and images. Second, most works (Liu & Chilton, 2022; Pavlichenko & Ustalov, 2023) apply similar modification to all prompts, lacking fine-grained, case-specific adjustments tailored to individual inputs, resulting in limited performance improvement. Third, existing methods exhibit limited generality (Cao et al., 2023; Rosenman et al., 2024), as they are typically designed for a single specific diffusion model. However, as illustrated in Figure 1b, different models have varying prompt preferences and may interpret the same prompt in different incorrect ways, making these methods difficult to extend to other models. These limitations highlight the importance of leveraging the outputs of generative models as model-specific feedback for effective optimization.

In this paper, we introduce VisualPrompter, a novel and training-free framework that provides fine-grained, case-specific, and model-preferred optimization for distinct user inputs. To make clear the optimization target, our model incorporates a SElf-REflection module (SERE) to identify the missing concepts that are included in the input prompts but not reflected in the generated images. SERE breaks down the prompts into atomic concepts using a large language model and verifies if each concept is reflected in the generated images using a Visual-Language Model (VLM). Subsequently, with the missing concepts as targets, a Target-Specific Prompt Optimization module (TSPO) is employed to refine the user prompts. By enriching the missing concepts while preserving the existing concepts, TSPO ensures that the refined prompts are both model-preferred and semantically faithful to the user's original intent. Furthermore, we introduce another LLM to incorporate appropriate aesthetic keywords into sentences, thereby improving the visual quality of optimized images.

Notably, by breaking down the user prompts into fine-grained concepts, our method allows case-specific detailed prompt optimization, which is not only more interpretable but also more powerful. This capability enables the model to assess key attributes like objects, properties, and relations, thereby refining and expanding vague or fragmented user inputs into well-structured, model-preferred prompts. Moreover, our method allows different prompt optimizations for different diffusion models according to the distinct answers based on their generated images. In this way, our VisualPrompter can be compatible with various generative models, obtaining a universal and robust performance improvement.

Experiments show that VisualPrompter outperforms prior prompt engineering works on two text-image alignment evaluation benchmarks, demonstrating the effectiveness of our approach. Images generated from texts optimized by VisualPrompter also achieve higher CLIP Score (Hessel et al., 2021) than those produced by previous prompt engineering methods. Besides this improved semantic fidelity, our optimization also enhances the aesthetic quality, contributing to more visually appealing results. Furthermore, our model exhibits notable performance improvements across various diffusion models, highlighting its extensive applicability and adaptability.

Our contributions are summarized as follows:

- We propose an innovative text-to-image prompt engineering method, named Visual-Prompter. Our approach is capable of generating prompts that effectively align with both model preferences and user intent.
- We propose a feedback-driven optimization system that leverages the visual feedback from the VLM to strategically enhance outputs from diverse generative models.

- Our method operates at the atomic semantic level to analyze and refine the prompt, fully preserving its original meaning while incorporating appropriate new content.
- Extensive experiments demonstrate the effectiveness of our VisualPrompter, which significantly outperforms current state-of-the-art prompt engineering methods in multiple benchmarks.

## 2 RELATED WORK

### 2.1 TEXT-TO-IMAGE SYNTHESIS

Diffusion models (Ho et al., 2020; Song et al., 2021) have recently attained unprecedented success in image synthesis. These models generate images from random noise through a series of denoising steps, achieving high visual fidelity and diversity. The introduction of classifier-free guidance (Ho & Salimans, 2022) facilitates the generation of images from brief text descriptions by leveraging text encoders such as CLIP (Radford et al., 2021), making text-to-image synthesis possible within the diffusion framework. Another significant innovation is the Latent Diffusion Model (LDM) (Rombach et al., 2022), which implements the diffusion process in latent space. LDM is capable of generating images with exceptional visual fidelity, while significantly reducing the required computational resources. Furthermore, the emergence of open-source diffusion models (Esser et al., 2024; Podell et al., 2024) has brought text-to-image generation into the public spotlight. Consequently, a well-designed prompt engineering framework can be immensely beneficial for novice users.

### 2.2 PROMPT ENGINEERING

Prompt engineering aims to optimize the interaction between humans and AI systems by crafting effective input prompts that elicit desired outputs (Liu et al., 2023; Sahoo et al., 2024). In the field of text-to-image generation, prompt engineering involves carefully selecting and combining phrases to achieve high-quality and satisfactory synthesized images. Early studies (Liu & Chilton, 2022; Pavlichenko & Ustalov, 2023) have focused on automatically searching for keywords that significantly influence the style and quality of generated images through mining techniques. With the rapid advancement of large language models, several studies (Cao et al., 2023; Wang et al., 2023a; Rosenman et al., 2024) have explored their utilization as prompt engineers, particularly through fine-tuning and reinforcement learning. These models are capable of producing detailed and varied content prompts, based on short phrases provided by users.

To achieve precise control, other methods (Brade et al., 2023; Feng et al., 2024) focus on the interaction with humans. PromptCharm (Wang et al., 2024b) provides convenient attention adjustments for the keywords within prompts. Prompt Expansion (Datta et al., 2024) outputs a set of expanded text prompts and images, providing users with multiple optimization directions for selection. Although these methods can improve the visual quality of generated images, they often neglect the semantic consistency with user-provided descriptions, which significantly limits their practicality in real-world scenarios.

## 3 VISUALPROMPTER

In this section, we present our VisualPrompter in detail, which is a semantic-aware, plug-and-play prompt engineering method for the text-to-image generation task.

### 3.1 OVERVIEW

The target of our VisualPrompter is to automatically optimize user input prompts into model-preferred prompts. The principal strength of our approach stems from its atomic semantic-level prompt optimization, which offers more precise and reliable control over semantics compared to direct expansion by models. Figure 2 illustrates the overall framework of our proposed method.

Given a user input prompt, a SElf-REflection module (SERE) is employed to identify the specific aspects that require enhancement. It analyzes the image generated by the diffusion model based on the input text. By thoroughly evaluating the semantic information in the prompt through question

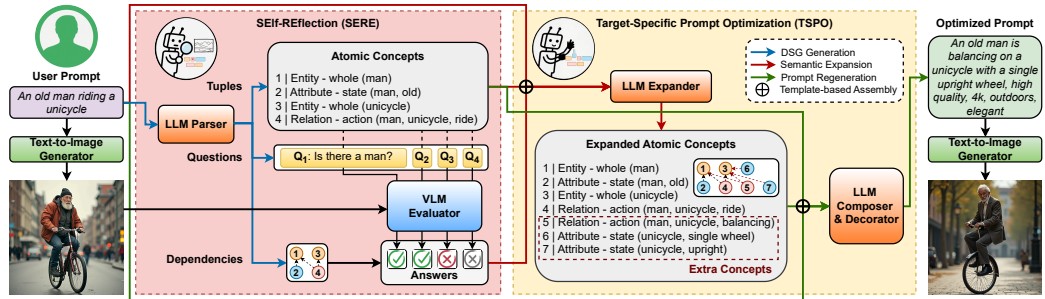

Figure 2: The refinement pipeline of our VisualPrompter. The optimization starts with LLM-driven VLM-verified semantic evaluation, followed by LLM-controlled concept expansion and sentence composition. Our approach emulates the chain-of-thought reasoning in human prompt refinement. All utilized models do not require any extra training.

generation and answering, we can effectively identify the missing concepts. Notably, the missing concepts are those textual descriptions that are included in user inputs but are not reflected in the generated images. To obtain prompts that remain faithful to user inputs while aligning with model preferences, we employ a Target-Specific Prompt Optimization module (TSPO) to refine the user prompt based on the identified missing concepts. TSPO first strategically enriches the attributes of the missing concepts to simulate model-preferred patterns. It then combines these expanded atomic concepts to form a syntactically complete and semantically fluent result.

## 3.2 SELF-REFLECTION MODULE

Our self-reflection module utilizes multi-step reasoning to obtain explicit visual feedback by evaluating the generated images against sentence-derived atomic concept questions. This mechanism ensures a fine-grained evaluation of the alignment between textual descriptions and visual content, effectively pinpointing discrepancies where the generative model fails to capture specific aspects of the user input. An example of this fine-grained evaluation is shown in Figure 2.

**Question Generation.** For detailed question generation, we adopt Davidsonian Scene Graph (DSG) (Cho et al., 2024) to identify atomic concepts within the input prompts. A DSG is a structured representation comprising a set of general questions. Each question corresponds to an atomic concept in the given prompt. These questions are interconnected through directed edges, which capture the entailment dependencies among them. We employ a Large Language Model (LLM) to generate the DSG, and its detailed construction process is elaborated in Appendix A.

**Question Answering.** Upon completing the construction of DSG, we employ a Visual-Language Model (VLM) as the visual question answering model to evaluate the generated image against the questions in DSG. The VLM is instructed to provide a binary response for each question, determining if the corresponding concept appears in the generated image. The dependency relationships within the DSG are utilized for question pruning, a process that eliminates the need for redundant assessments. Specifically, if a particular concept is judged to be missing, all subsequent concepts dependent on it are automatically regarded as missing as well. The answers enable a comprehensive and fine-grained semantic analysis of the generated images, which will be employed as semantic-aware information to refine the prompt.

## 3.3 TARGET-SPECIFIC PROMPT OPTIMIZATION

With these identified missing concepts by our self-reflection module, we can perform targeted optimization of the input texts, reducing unnecessary modifications and expansions. We introduce a novel, training-free optimization approach that effectively manages prompt adjustments.

**Prompt Regeneration.** Rather than directly rewriting the sentences, VisualPrompter first utilizes an LLM to expand the generated atomic concepts based on the responses to the questions from the SERE module. This expansion process specifically targets the identified missing concepts, enriching them with additional detail such as attributes, actions, and spatial relationships. Our observations

reveal that augmenting the missing concepts can nudge the user input toward the distribution of model-preferred prompts, providing a plug-and-play lever for optimizing various generative models. These additional elements are also expressed as atomic concepts, ensuring consistency and clarity in representation. Notably, if no missing concepts are detected, no further modifications will be made.

Subsequently, VisualPrompter generates a new prompt from the expanded atomic concepts with an LLM. The original prompts and concepts are also provided to guide the synthesis, ensuring that the generated prompt remains contextually aligned with the initial input. The TSPO module retains textual descriptions that are already accurately reflected in the generated images to avoid unnecessary modifications, thereby preserving original user intent. By adopting this two-step optimization methodology, our framework is able to produce more accurate and reliable prompts.

**Prompt Decoration.** To further enrich the diversity and aesthetic quality of the generated content, we augment the optimized results with several aesthetic keywords that are automatically selected by an LLM. By providing example keywords such as *high-quality*, *detailed*, and *fantasy*, we leverage the LLM to automatically generate rich and diverse keywords that do not conflict with the input sentences. These stylistic keywords are seamlessly integrated into the texts or appended at the end in a comma-separated format.

## 4 EXPERIMENTS

In this section, we first present the benchmarks involved and our experimental configurations. Then, we compare VisualPrompter with three previous SOTA prompt engineering works both quantitatively and qualitatively. Human evaluation is conducted to underscore the preeminence of our approach. Furthermore, we select several widely-used text-to-image generation applications to demonstrate the effectiveness of our model in open-world scenarios.

### 4.1 BENCHMARKS AND EVALUATION METRICS

**Benchmarks.** For our evaluation, we utilize two comprehensive benchmarks: DSG-1k (Cho et al., 2024) and TIFA v1.0 (Hu et al., 2023). Both of them contain human-written prompts from various established text-to-image benchmarks. The DSG-1k benchmark comprises $1,060$ prompts from 10 publicly available prompt datasets, encompassing a wide range of skills and writing styles. It provides $8,182$ questions associated with these prompts, facilitating a multifaceted evaluation of generated images. Meanwhile, TIFA v1.0 contains $4,081$ diverse prompts from $4$ datasets. As the original TIFA v1.0 benchmark contains multiple-choice and free-form questions, we further refine these questions and represent them in the DSG structure. Ultimately, the TIFA benchmark used in our experiments contains $19,233$ general questions.

**Metrics.** To evaluate the text-image alignment, we adopt a robust and innovative approach (Yarom et al., 2023; Cho et al., 2023), which leverages question generation and answering techniques (Deutsch et al., 2021; Min et al., 2023). This VLM-as-judge metric demonstrates strong correlations with human judgments (Hu et al., 2023), making it particularly suitable for fine-grained evaluation tasks. We also employ the CLIP Score (Hessel et al., 2021) to evaluate the semantic alignment between textual descriptions and generated images at the feature level. For aesthetic assessment, we employ the Aesthetic Score (Schuhmann et al., 2022), a well-established metric for objective image quality assessment, reducing subjectivity in human judgments.

### 4.2 EXPERIMENTAL SETTINGS

Our VisualPrompter leverages a dual-model architecture that requires an LLM for DSG generation and prompt optimization, connected by a VLM for comprehensive image semantic evaluation. We employ the state-of-the-art open-source Qwen2 (Yang et al., 2024) as our primary language model and the Qwen2-VL (Wang et al., 2024a) as our visual question answering model. Each language generation task in our pipeline is initiated by feeding Qwen2 with a meticulously designed preamble and 23 manually curated samples from the TIFA dataset, ensuring consistent model output.

To thoroughly evaluate our framework's performance, we conduct extensive testing across three prominent diffusion models and one autoregressive generative model, including Stable Diffusion v1.5, Stable Diffusion v2.1 (Rombach et al., 2022), Flux-dev (BlackForestLabs, 2024), and Janus-

Table 1: Summarized results on the DSG and TIFA benchmark. Reported scores are based on the percentage of "yes" answers to the questions, while the best scores are highlighted in boldface. Our VisualPrompter demonstrates notable improvements in semantic consistency evaluations.

| Methods | DSG benchmark | | | | TIFA benchmark | | | | Average |
|---|---|---|---|---|---|---|---|---|---|
| | SD 1.5 | SD 2.1 | Flux-dev | Janus-Pro | SD 1.5 | SD 2.1 | Flux-dev | Janus-Pro | |
| Baseline | 67.5 | 72.1 | 79.1 | 79.5 | 75.0 | 80.4 | 87.9 | 85.7 | 78.4 |
| NeuroPrompts | 58.4 | 68.7 | 75.5 | 81.7 | 62.8 | 75.5 | 84.0 | 89.8 | 74.6 |
| Promptist | 65.1 | 68.7 | 76.9 | 78.0 | 71.4 | 77.2 | 85.3 | 86.9 | 76.2 |
| BeautifulPrompt | 47.9 | 49.6 | 51.5 | 55.3 | 50.9 | 53.6 | 57.4 | 62.0 | 53.5 |
| VisualPrompter | **69.5** | **77.0** | **84.3** | **82.6** | **80.7** | **82.8** | **93.8** | **93.7** | **83.0** |

Table 2: Semantic evaluation across all categories of the DSG benchmark using Flux-dev.

| Model | Category of source texts | | | | | | | | | |
|---|---|---|---|---|---|---|---|---|---|---|
| | whoops | localized | posescript | vrd | countbench | midjourney | tifa160 | draw text | stanford | diffusion db |
| Flux | 82.2 | 85.1 | 73.3 | 85.8 | 72.1 | 56.7 | 88.6 | 83.3 | 89.1 | 68.9 |
| NeuroPrompts | 75.3 | 86.1 | 74.8 | 84.8 | 70.3 | 49.6 | 85.7 | 75.6 | 88.2 | 58.5 |
| Promptist | 79.1 | 84.1 | 75.4 | 81.9 | 73.3 | 56.6 | 85.4 | 81.2 | 85.2 | 61.4 |
| BeautifulPrompt | 56.4 | 53.0 | 61.2 | 64.5 | 44.7 | 38.6 | 58.2 | 40.1 | 45.7 | 48.7 |
| VisualPrompter | **86.5** (+4.3) | **89.7** (+3.6) | **75.9** (+0.5) | **92.4** (+6.6) | **81.7** (+8.4) | **65.7** (+9.0) | **93.1** (+4.5) | **88.1** (+4.8) | **92.8** (+3.7) | **72.1** (+3.2) |

Pro (Chen et al., 2025). Our evaluation utilizes prompts from both the DSG-1k and TIFA benchmarks. We compare our method with several open-source prompt engineering methods, including NeuroPrompts (Rosenman et al., 2024), Promptist (Hao et al., 2023), and BeautifulPrompt (Cao et al., 2023). All of these methods utilize supervised fine-tuning and reinforcement learning to adjust LLMs for optimizing prompts. Our approach is implemented using PyTorch. All experiments are conducted on a single A100 with 40GB memory.

### 4.3 SEMANTIC ACCURACY EVALUATION

We compare VisualPrompter with several state-of-the-art prompt engineering methods on two semantic consistency evaluation benchmarks, and summarize the results in Table 1. We refine the original text prompts with each method, and use the refined prompts to generate images with different generative models. We then use Qwen2-VL-7B to assess whether the generated images contain the specified semantic information. We quantify the results by measuring the average answering accuracy, which has been demonstrated to be effective by DSG (Cho et al., 2024).

As illustrated in Table 1, our model achieves significant improvements across all evaluation benchmarks and all generative models. Notably, VisualPrompter consistently outperforms the four baseline generative models in terms of average performance scores. The results also show that VisualPrompter achieves higher performance gains when integrated with Flux-dev and Janus-Pro, which can be attributed to their state-of-the-art generative capabilities, particularly in handling long sentences and intricate details. This indicates that VisualPrompter can better collaborate with high-capacity generators to produce results with more accurate semantics. It also demonstrates the strong adaptability of VisualPrompter, enabling its effective application across various scenarios.

Surprisingly, previous methods not only fail to preserve the semantic consistency during optimization but also exhibit varying degrees of decline in this aspect. Our analysis reveals that both NeuroPrompts and Promptist are unable to modify the original descriptions, even though they utilize an LLM for prompt optimization. This limitation prevents them from effectively refining those model-dispreferred sentences. Additionally, they often introduce irrelevant keywords that are inconsistent with the overall context of the prompts, thereby compromising the semantic fidelity of the generated images. In contrast, BeautifulPrompt is able to adjust the original prompts and add contextually relevant details. However, it tends to omit critical information during the modification process, which significantly deviates from the user intent.

In Table 2, we present the semantic evaluation results across all categories of the DSG benchmark using Flux-Dev. Additional results from other generative models, as well as those from the TIFA benchmark, can be found in Appendix E. These results confirm that our approach effectively rein-

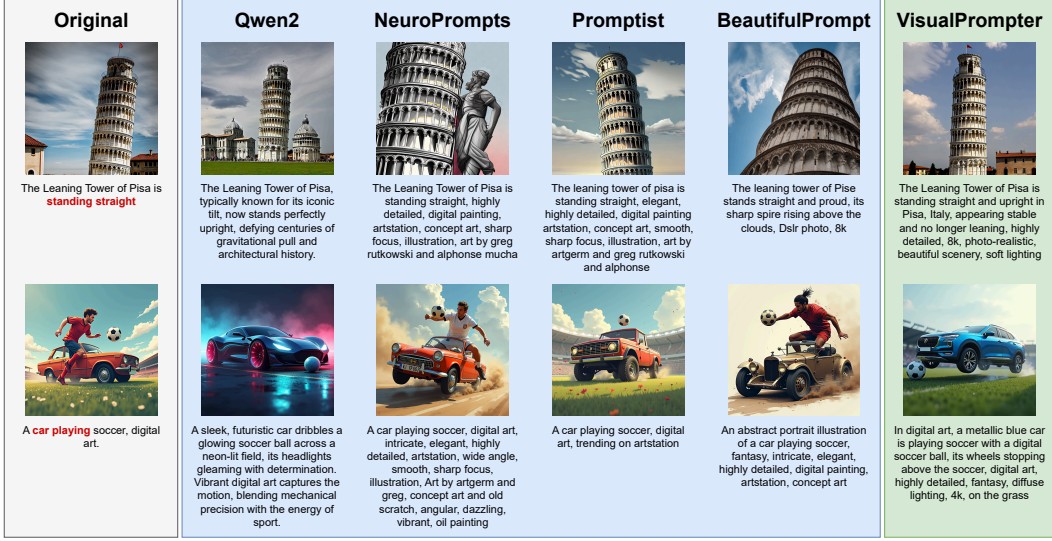

Figure 3: Comparison of prompts generated by large language model and different prompt engineering methods. Images in the two rows correspond to outputs from Stable Diffusion v2.1 and Flux-dev, respectively.

forces semantic components in the input prompts and ensures their accurate representation in the generated images. Our model demonstrates consistent improvements across specific categories, indicating its strong capability to comprehend and enhance key elements such as objects, attributes, and relationships.

We visualize the generation results of various generative models, as shown in Figure 3. We also use Qwen2 for zero-shot prompt optimization as a comparison. The results demonstrate that our model is capable of producing more elegant prompts that result in more visually compelling images. It successfully corrects the deficiencies of the original images, indicating that our optimized prompts are preferred by generative models. Additional visual examples are presented in Appendix F.

## 4.4 TEXT-IMAGE RELEVENCE EVALUATION

To further validate the effectiveness of Visual-Prompter, we evaluate the CLIP Score between the optimized prompts and their corresponding generated images. We report the mean scores of all the prompts from the DSG benchmark, generating images with those different generative models. The results are illustrated in Table 3.

As shown in Table 3, prompts optimized by VisualPrompter achieve the highest scores across all the generative models. The results indicate that the prompts produced by VisualPrompter can be effectively understood by the generative models, generating images with improved semantic coherence. Optimized prompts from Promptist and BeautifulPrompt also achieve a relatively high CLIP Score compared to baselines, whereas NeuroPrompts performs the worst. This is primarily because NeuroPrompts offers only limited keyword options, which significantly restricts the diversity of the resulting prompts. In contrast, we leverage LLMs to generate aesthetic keywords that align well with the sentence content, leading to greater overall coherence and harmony in the prompts.

Table 3: Results of CLIP Score between optimized prompts and generated images on DSG benchmark. Baseline represents the results on images from original prompts.

| Methods | Generative Models | | | | Mean |
|---|---|---|---|---|---|
| | SD v1.5 | SD v2.1 | Flux | Janus-Pro | |
| Baseline | 31.53 | 31.88 | 31.58 | 31.83 | 31.71 |
| NeuroPrompts | 25.61 | 25.26 | 24.56 | 26.21 | 25.41 |
| Promptist | 32.14 | 31.61 | 31.49 | 31.85 | 31.77 |
| BeautifulPrompt | 31.70 | 31.38 | 31.35 | 32.10 | 31.63 |
| VisualPrompter | **32.21** | **32.50** | **33.81** | **32.22** | **32.69** |

Additionally, we also notice that Promptist and BeautifulPrompt achieve higher CLIP Score than the baseline with Stable Diffusion v1.5 and Janus-Pro, while their performance drops with Stable

Table 4: Results of Aesthetic Score for generated images on DSG benchmark. Baseline represents the results on images from original prompts.

| Methods | Generative Models | | | Mean |
| | SD v1.5 | SD v2.1 | Flux | |
|---|---|---|---|---|
| Baseline | 5.30 | 5.48 | 5.77 | 5.52 |
| NeuroPrompts | **6.21** | 6.03 | 6.40 | 6.21 |
| Promptist | 5.93 | 5.86 | 6.25 | 6.01 |
| BeautifulPrompt | 5.97 | 5.98 | **6.48** | 6.14 |
| VisualPrompter | 5.66 | 5.73 | 6.03 | 5.81 |
| Ours + NeuroPrompts | 6.13 | **6.15** | 6.44 | **6.24** |

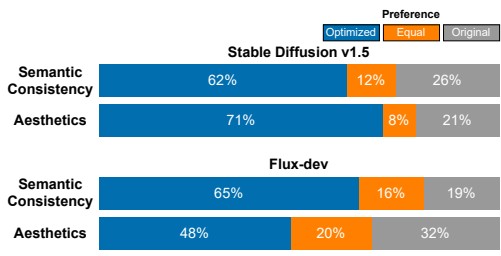

Figure 4: Human evaluation results. From left to right, the segments indicate preference for images from optimized prompts, equal preference for both results, and preference for images from original prompts.

Diffusion v2.1 and Flux-dev. The performance gap arises from their specific optimization for Stable Diffusion v1.5. Another reason is that Janus-Pro's autoregressive design enables stronger semantic understanding, allowing it to better handle complex sentences. The discrepancy highlights their limited generalization capability across different generative models. In contrast, our prompt engineering framework functions in a model-agnostic way, which does not rely on any specific generative model, demonstrating the broad compatibility and robustness of VisualPrompter.

## 4.5 AESTHETIC EVALUATION

Recent studies (Hao et al., 2023; Podell et al., 2024) have revealed that conventional image quality metrics, such as FID (Heusel et al., 2017) and CLIP Score (Hessel et al., 2021), do not exhibit a positive correlation with visual aesthetics. To alleviate the burden of labor-intensive human evaluation, the Aesthetic Score (Podell et al., 2024) has emerged as an effective metric, as it is specifically designed to align with human perceptions of beauty, leveraging insights from large-scale datasets. Thus, previous studies have placed greater emphasis on this metric and integrated it into their reinforcement learning frameworks. We also evaluate our method with the Aesthetic Score to show the visual elegance of optimized images. The results are summarized in Table 4.

The experimental results on aesthetics show that VisualPrompter achieves modest improvements in aesthetic quality, which are not as substantial as those in previous studies. This discrepancy primarily stems from the tendency of our method to classify the original prompts as descriptions of realistic photographs, which results in an inherent lack of imaginative or fantastical elements among the optimized prompts. This also indicates that our prompt decorator is overly simplistic for enhancing aesthetics. To address this problem, we employ NeuroPrompts as a replacement for our decorator, which adds additional stylistic keywords while preserving the original sentence. This integrated approach achieves optimal performance on average in the aesthetic evaluation. However, we observe a notable decline in semantic consistency, compared to the results without the involvement of NeuroPrompts. It implies that adopting the Aesthetic Score as a training metric may inadvertently encourage the model to optimize for superficial features at the expense of semantic integrity, highlighting the need for more balanced and semantically meaningful training guidance.

## 4.6 USER STUDY

To enhance the credibility of the above quantitative results, we additionally conduct a human evaluation experiment based on Stable Diffusion v1.5 and Flux-dev. We compute the average preference distribution and report the results in Figure 4. In detail, we generate image pairs using both the original prompts and the optimized prompts derived from VisualPrompter. A panel of 10 independent annotators is tasked with selecting the most desirable images from semantic and aesthetic aspects. Each annotator is provided with 50 samples randomly selected from both benchmarks.

The results of human evaluation indicate that annotators generally prefer images generated by optimized prompts in both semantic and aesthetic aspects, which demonstrates the superiority of our

Table 5: Impact of different modules in Visual-Prompter. We report the Semantic Accuracy / Aesthetic Score based on the DSG benchmark.

| Visual Feedback | Prompt Modification Approach | | |
|---|---|---|---|
| | None | Qwen | DSG |
| w/o | 72.1 / 5.48 | 67.8 / 5.32 | 71.9 / 5.69 |
| w/ | 73.8 / 5.54 | 73.0 / 5.45 | **77.0 / 5.73** |

Table 6: Inference time comparison.

| T2I Models | Prompt Modification Approach | | | |
|---|---|---|---|---|
| | NeuroPrompts | Promptist | Qwen3-32B | Ours |
| SD v1.5 | 8.5s | 7.0s | 21.3s | 10.7s |
| FLUX-dev | 20.3s | 18.8s | 33.1s | 34.3s |

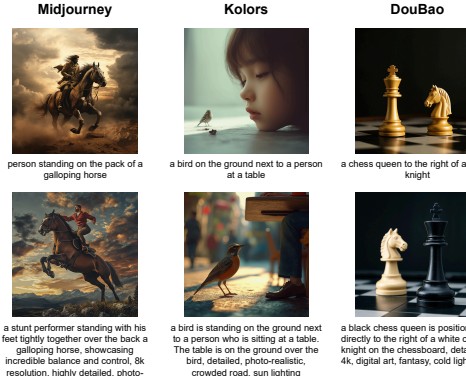

Figure 5: Adaptation for online generators. The second row presents results optimized by VisualPrompter.

approach. The optimized prompts slightly outperform the original prompts in terms of aesthetics on Flux-dev, owing to its robust generative capability, which can effectively fill in intricate details.

### 4.7 ABLATION STUDY

**Module Impact Evaluation.** To systematically analyze the impact of different components in VisualPrompter, we conduct an ablation study to evaluate the individual contributions of the self-reflection module and the target-specific optimization module in Figure 2. Specifically, we compare two approaches for prompt refinement: direct prompt optimization using Qwen and constructing prompts from expanded concepts based on DSG. For the self-reflection module, visual feedback is leveraged to refine the targeted location of the given prompts. Additionally, we account for the impact of regeneration, where visual feedback is employed to evaluate the original images. Prompts that result in defective images will be used to regenerate images without any modifications.

As shown in Table 5, our fine-grained semantic extraction and composition approach achieves better results than directly using LLM for optimization. We observe that Qwen often produces prompts with semantically irrelevant details. These long sentences increase the difficulty for the diffusion models to comprehend them, leading to worse results than the original prompts. Another observation is that the regeneration of images actually exerts a non-negligible impact on the semantic evaluation metric. Despite this influence, our VisualPrompter achieves superior results, demonstrating its outstanding capability in enhancing semantic consistency.

**Inference time.** We compare the inference times of several optimization methods and summarize the results in Table 6. We calculate the total time from when a prompt is provided to when a complete image is generated. Our model uses Qwen2 1.5 B for fast DSG generation and concept integration. As shown in Table 6, while our VisualPrompter is marginally slower compared with existing optimization models, the difference falls within tolerable limits for practical applications.

**Online Generation Evaluation.** To show the robust adaptability of VisualPrompter, we select several online generative models, including Midjourney (Midjourney, 2021), Kolors (Kolors, 2024), and DouBao (DouBao, 2023). We provide them with human-written prompts and utilize the feedback to further optimize the prompts. The results are presented in Figure 5. Each image is selected from three samples with the same prompts.

## 5 CONCLUSION

In this paper, we propose VisualPrompter, a training-free prompt engineering framework that improves text-to-image generation by refining user prompts to model-preferred sentences. We reveal that prompts generated by existing LLM-based methods often struggle to maintain semantic consistency with the original intent and consequently produce images that deviate semantically. To tackle

this issue, VisualPrompter operates at the atomic semantic level to analyze and revise the prompt. It leverages model-specific insights obtained through visual reflection to perform targeted expansion and modification of the initial input, thereby ensuring semantic consistency throughout the optimization process. Extensive experiments demonstrate the effectiveness of our VisualPrompter, which achieves state-of-the-art performance in enhancing semantic consistency while maintaining superior aesthetic quality. Compared with previous methods, VisualPrompter exhibits better generalization capabilities, which can be easily adapted to various diffusion models. We intend to extend this feedback-driven optimization system to multimodal generative models in our future work.

## ACKNOWLEDGEMENTS

This research is supported by Artificial Intelligence-National Science and Technology Major Project (2023ZD0121200), and the National Natural Science Foundation of China (62531026, 62437001, 62436001), and the Natural Science Foundation of Jiangsu Province under Grant BK20243051, and the Strategic Priority Research Program of Chinese Academy of Sciences under Grant XDA04080400.

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

## A  AUTOMATIC CONSTRUCTION OF DSG

Davidsonian Scene Graph (DSG) (Cho et al., 2024) is a set of questions in the form of a directed acyclic graph, which has been used for text-image alignment evaluation. As shown on the right side of Figure 6, each question in DSG is about an atomic concept, which represents a specific and indivisible semantic unit through a tuple. By feeding the generated images and the related questions into a visual question answering model, we can effectively identify and analyze the missing concepts present in the images.

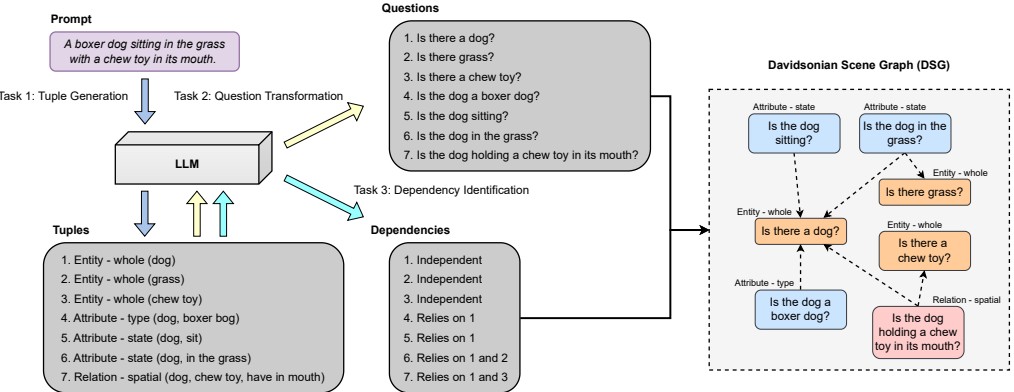

Figure 6: The automatic construction of DSG with an LLM.

Large language models (LLMs) have been employed to construct DSG from textual descriptions automatically. To meticulously construct a DSG that is both comprehensive and non-redundant, our approach adopts a three-step pipeline for the overall generation process, as illustrated in Figure 6. Specifically, an input prompt is first decomposed into detailed concepts represented as tuples, which include entities, attributes, relations, and actions in the text. Based on these tuples of atomic concepts, LLMs are able to generate corresponding general interrogative sentences and determine the dependency relationships among them. Finally, the questions and dependencies are combined to form the DSG for further use.

To ensure consistent and standardized outputs from the large language model (LLM), for each step in the process, we provide the LLM with a well-designed preamble and multiple manually annotated samples. This methodology guides the LLM to generate tuple annotations in a uniform format for new inputs, thereby enhancing the reliability and scalability of DSG generation across diverse textual inputs. We assign a numerical identifier to each tuple to distinguish between them and ensure the integrity of data. An example of tuple generation is shown in Figure 7. The preambles and annotated examples for other tasks will be included in our code release.

For the generation of questions and dependencies, we provide the LLM with both the prompts and tuples to enhance comprehension. Any invalid results will be regenerated multiple times to ensure accuracy.

## B  TARGET-SPECIFIC OPTIMIZATION

Here we provide examples for the two key technologies in our target-specific optimization module: concept expansion and prompt regeneration. We utilize the LLM engineering method used in DSG generation above. Examples of prompt templates are shown in Figure 8 and Figure 9.

## C  KEYWORDS FOR DECORATION

To enhance the aesthetics of generated images, we employ an LLM decorator to incorporate keywords, which have been demonstrated to substantially improve the aesthetic appeal of the images. To constrain the generation scope of the LLM, we provide several keyword examples in the task

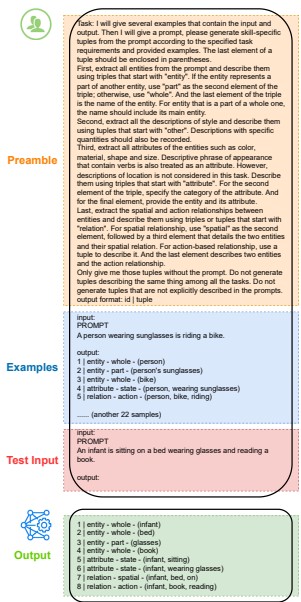

Figure 7: Example of tuple generation.

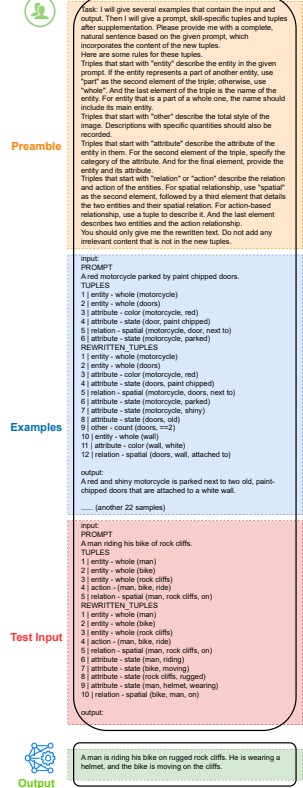

Figure 8: Example of prompt regeneration.

Figure 9: Example of concepts expansion.

preamble and instruct the LLM decorator to select appropriate keywords that align with the prompt content. These suitable keywords are appended at the end of the input sentence, separated by commas. We collect some keywords from the results of NeuroPrompts and BeautifulPrompt. These

Table 7: Class and examples of aesthetic keywords.

| class | Samples |
|---|---|
| quality | best quality, 4k, 8k, highres, masterpiece, fantasy art, highly detailed |
| style | photo-realistic, oil painting, portraits, digital art, landscape, impressionist, anime, concept artists, cyberpunk |
| background | blue sky, crowded road, beautifully decorated wall, high mountain |
| light | studio lighting, soft lighting, sun lighting, diffuse lighting, cold lighting |
| aesthetics | beautiful, elegant, stunning, lovely, majestic, charming, graceful |

Table 8: Detailed results on the TIFA benchmark. Reported scores represent the percentage of "yes" answers.

| Model | coco | Category of source texts | | paintskill | Average |
|---|---|---|---|---|---|
| | | drawbench | partiprompt | | |
| SD v1.5 | 81.8 | 65.2 | 72.5 | 58.5 | 75.0 |
| NeuroPrompts | 69.7 | 49.5 | 58.2 | 52.4 | 62.8 |
| Promptist | 78.1 | 62.4 | 67.9 | 57.8 | 71.4 |
| BeautifulPrompt | 56.3 | 50.8 | 45.9 | 43.1 | 50.9 |
| VisualPrompter | 87.5 (+5.7) | 70.5 (+5.3) | 77.0 (+4.5) | 66.8 (+8.3) | 80.7 (+5.7) |

(a) Semantic evaluation on Stable Diffusion v1.5.

| Model | coco | Category of source texts | | paintskill | Average |
|---|---|---|---|---|---|
| | | drawbench | partiprompt | | |
| SD v2.1 | 86.2 | 69.1 | 77.9 | 67.5 | 80.4 |
| NeuroPrompts | 81.4 | 64.5 | 72.5 | 64.3 | 75.5 |
| Promptist | 82.3 | 67.1 | 74.8 | 67.3 | 77.2 |
| BeautifulPrompt | 58.9 | 50.3 | 48.9 | 46.8 | 53.6 |
| VisualPrompter | 89.0 (+2.8) | 74.5 (+5.4) | 79.6 (+1.7) | 69.7 (+2.2) | 82.8 (+2.4) |

(b) Semantic evaluation on Stable Diffusion v2.1.

| Model | coco | Category of source texts | | paintskill | Average |
|---|---|---|---|---|---|
| | | drawbench | partiprompt | | |
| Flux | 92.4 | 80.8 | 86.2 | 77.2 | 87.9 |
| NeuroPrompts | 88.6 | 74.2 | 80.3 | 79.2 | 84.0 |
| Promptist | 90.0 | 75.6 | 82.3 | 78.2 | 85.3 |
| BeautifulPrompt | 63.2 | 56.1 | 49.8 | 56.3 | 57.4 |
| VisualPrompter | 96.2 (+3.8) | 87.6 (+6.8) | 90.8 (+4.6) | 94.7 (+5.5) | 93.8 (+5.9) |

(c) Semantic evaluation on Flux-dev.

| Model | coco | Category of source texts | | paintskill | Average |
|---|---|---|---|---|---|
| | | drawbench | partiprompt | | |
| Janus-Pro | 90.8 | 85.4 | 80.2 | 81.1 | 85.7 |
| NeuroPrompts | 92.9 | 85.2 | 87.9 | 83.9 | 89.7 |
| Promptist | 90.4 | 79.4 | 84.8 | 81.1 | 86.9 |
| BeautifulPrompt | 67.1 | 60.3 | 56.6 | 57.4 | 62.0 |
| VisualPrompter | 95.4 (+2.5) | 88.2 (+2.8) | 91.9 (+4.0) | 93.8 (+9.9) | 93.7 (+4.0) |

(d) Semantic evaluation on Janus-Pro.

keywords are systematically organized according to their descriptive categories, including image quality, artistic style, background description, environmental illumination, and perspective. A selection of examples is illustrated in Table 7.

For each class of keywords, we guide the LLM to select or create 0 to 2 words and incorporate them into the input prompts. This approach yields sentences that are both detailed and enriched with aesthetic keywords. Our experimental findings indicate that the inclusion of these aesthetic keywords does not substantially affect the semantic consistency between generated images and textual descriptions.

## D   EVALUATION RELIABILITY WITH DSG

We adopt the Davidsonian Scene Graph (DSG) (Cho et al., 2024) as a stable metric for text-image alignment. DSG decomposes prompts into atomic, dependency-structured questions whose accuracy yields a precise measurement of semantic consistency. The alignment score is computed as the proportion of "yes" responses from a VLM, ensuring that images with stronger textual fidelity receive higher scores. Notably, DSG demonstrates near-perfect reliability: questions achieve 98.3% precision and 96.0% recall against human annotations, while state-of-the-art VLMs answer these questions with 82.6% accuracy in entity judgments. These results substantiate the reliability of DSG, enabling our VisualPrompter to deliver dependable defect detection and targeted optimization. Furthermore, DSG's dependency-aware design inherently prunes invalid queries such as asking about an attribute when the object is absent, thereby reducing noise and reinforcing measurement stability across diverse prompts and models. For additional experiments and implementation details validating DSG's stability, please refer to Cho et al. (2024).

## E   DETAILED RESULTS ON BENCHMARKS

The benchmarks used in our study, namely TIFA v1.0 (Hu et al., 2023) and DSG (Cho et al., 2024) (as introduced in the paper), consist of multiple constituent datasets. We have systematically evaluated VisualPrompter's performance across all sub-datasets and compiled the comprehensive results in Table 8 and Table 9.

Table 9: Detailed results on the DSG benchmark. The best scores are highlighted in boldface. Our VisualPrompter stands out as the only method that demonstrates notable improvement in semantic consistency evaluation.

| Model | whoops | localized | posescript | vrd | Category of source texts | | tifa160 | draw text | stanford | diffusion db | Average |
|---|---|---|---|---|---|---|---|---|---|---|---|
| | | | | | countbench | midjourney | | | | | |
| SD v1.5 | 69.8 | 70.9 | **65.9** | 66.7 | 68.0 | 61.5 | 77.1 | **55.2** | **65.6** | 68.6 | 67.5 |
| NeuroPrompts | 57.8 | 62.9 | 58.3 | 62.2 | 54.1 | 50.3 | 64.6 | 47.6 | 62.0 | 60.6 | 58.4 |
| Promptist | 73.7 | 68.0 | 61.8 | 64.7 | 66.2 | 57.9 | 70.5 | 56.1 | 65.1 | 63.3 | 65.1 |
| BeautifulPrompt | 48.0 | 48.3 | 52.8 | 53.7 | 43.7 | 46.3 | 50.1 | 41.5 | 38.9 | 54.5 | 47.9 |
| VisualPrompter | **76.5** (+2.8) | **75.7** (+4.8) | 61.5 (-4.4) | **71.4** (+4.7) | **72.3** (+4.3) | **63.4** (+1.9) | **77.8** (+0.7) | 54.5 (-1.6) | 63.6 (-2.0) | **73.2** (+4.6) | **69.5** (+2.0) |

(a) Semantic evaluation on Stable Diffusion v1.5.

| Model | whoops | localized | posescript | vrd | Category of source texts | | tifa160 | draw text | stanford | diffusion db | Average |
|---|---|---|---|---|---|---|---|---|---|---|---|
| | | | | | countbench | midjourney | | | | | |
| SD v2.1 | 76.9 | 77.7 | 66.9 | 70.6 | 69.5 | 63.0 | 81.6 | 64.1 | 73.3 | 71.6 | 72.1 |
| NeuroPrompts | 70.8 | 76.4 | 62.2 | 74.5 | 66.4 | 58.7 | 74.5 | 63.8 | 66.1 | 70.1 | 68.7 |
| Promptist | 71.5 | 73.3 | 65.8 | 68.1 | 71.2 | 59.8 | 75.9 | 67.1 | 64.6 | 65.7 | 68.7 |
| BeautifulPrompt | 54.7 | 47.3 | 55.2 | 56.4 | 47.1 | 44.6 | 52.4 | 41.3 | 43.4 | 52.1 | 49.6 |
| VisualPrompter | **82.0** (+5.1) | **80.6** (+2.9) | **71.1** (+4.2) | **77.5** (+3.0) | **79.1** (+9.6) | **69.6** (+6.6) | **83.8** (+2.2) | **73.3** (+6.2) | **74.2** (+0.9) | **74.9** (+3.3) | **77.0** (+4.9) |

(b) Semantic evaluation on Stable Diffusion v2.1.

| Model | whoops | localized | posescript | vrd | Category of source texts | | tifa160 | draw text | stanford | diffusion db | Average |
|---|---|---|---|---|---|---|---|---|---|---|---|
| | | | | | countbench | midjourney | | | | | |
| Janus-Pro | 86.1 | 87.2 | 69.8 | 82.4 | 73.7 | 65.6 | 87.2 | 79.9 | 85.9 | 72.1 | 79.5 |
| NeuroPrompts | 86.2 | **88.7** | **76.7** | 86.4 | 76.0 | 68.3 | 88.1 | **85.9** | **86.8** | 70.0 | 81.7 |
| Promptist | 84.6 | 80.4 | 68.1 | 80.3 | 76.2 | 65.6 | 88.3 | 78.9 | 82.5 | 69.2 | 78.0 |
| BeautifulPrompt | 66.3 | 54.2 | 59.1 | 62.4 | 54.7 | 45.2 | 61.0 | 45.5 | 44.3 | 56.8 | 55.3 |
| VisualPrompter | **91.2** (+5.0) | 86.6 (-2.1) | 74.2 (-2.5) | **86.9** (+0.5) | **79.0** (+2.8) | **69.8** (+1.5) | **90.8** (+2.5) | 85.8 (-0.1) | 80.1 (-6.7) | **76.9** (+4.8) | **82.6** (+0.9) |

(c) Semantic evaluation on Janus-Pro.

From Table 9, our model demonstrates more pronounced improvements across prompts from *whoops*, *localized*, and *countbench* datasets, which care more about object attributes and spatial relationships. The results demonstrate that our model is capable of effectively enhancing semantic elements in the prompt, ensuring their representation in the generated images. We also notice that there are still 3 sub-datasets in the detailed tables on which our method exhibits a slight performance decline. We attribute this fact to the limitation of Stable Diffusion v1.5, which is not sufficiently capable of comprehending human poses and processing long textual descriptions.

The experimental results demonstrate that our method achieves significant improvements across all subcategories of the TIFA benchmark, indicating its strong capability in optimizing naturally described sentences. On the DSG benchmark, our model shows consistent performance gains in most subcategories, which further validates its generalization ability across different evaluation metrics.

While our approach delivers robust performance overall, we observe marginal performance decreases when applied to SD v1.5 and Janus-Pro models on the DSG benchmark. This can be attributed to two main factors: (1) SD v1.5's limited capacity in processing moderately complex sentences often leads to substantial detail loss, and (2) Janus-Pro's superior language understanding capability makes it particularly responsive to NeuroPrompt-optimized sentences, yielding slightly higher scores. We plan to conduct more in-depth investigations to develop better optimization strategies in our future work.

## F  MORE CASES OF VISUALPROMPTER

Additional results for our optimization approach are shown in Figure 10, Figure 11, and Figure 12. Each image pair consists of one generated from the original prompt and another from the prompt optimized by our VisualPrompter, with the corresponding prompt text displayed beneath each image. The visualization results convincingly demonstrate that our VisualPrompter exhibits robust adaptability to various prompts and substantiates superior performance in rectifying multiple types of generation artifacts.

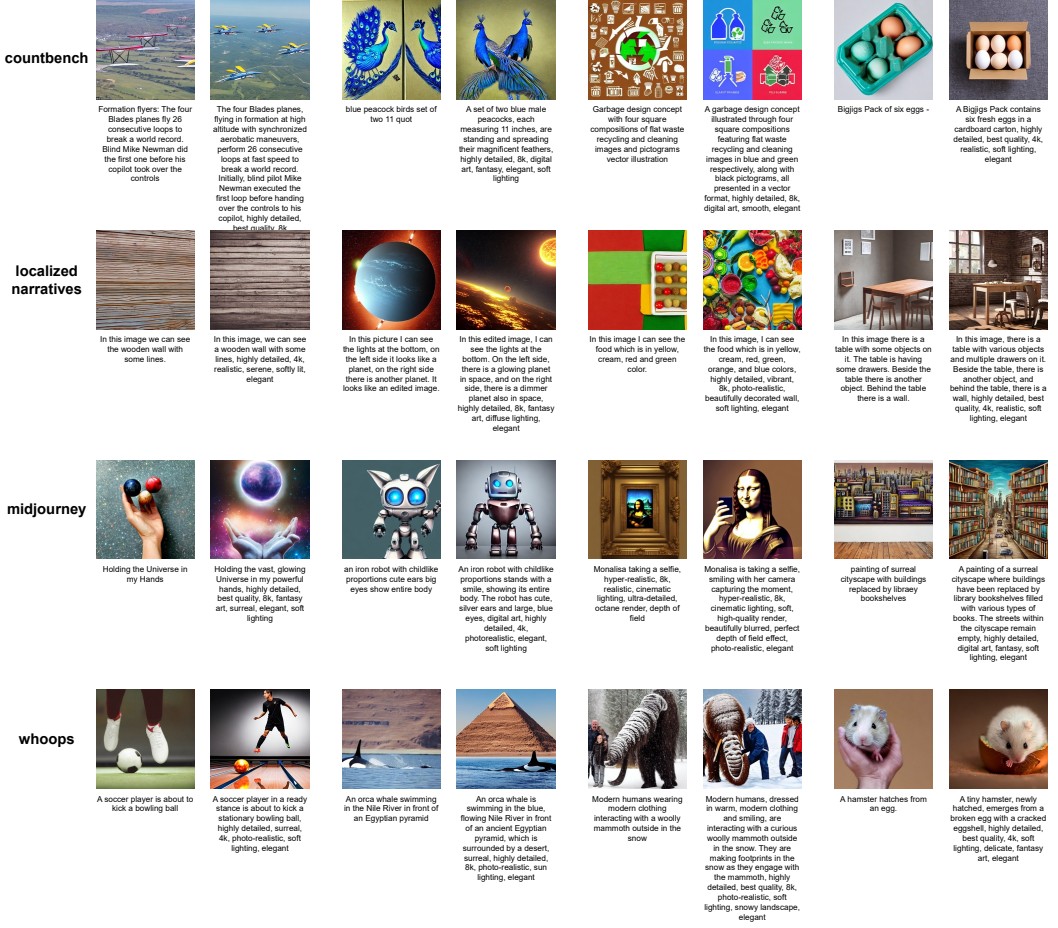

Figure 10: Examples of images generated by Stable Diffusion v1.5. Each line corresponds to the results of prompts derived from distinct sub-datasets. The right image in each pair is generated from the prompt optimized by VisualPrompter, as detailed below the image.

## G    FAILURE CASES OF VISUALPROMPTER

Despite its effectiveness, our VisualPrompter still exhibits certain limitations. A few typical failure cases are shown in Figure 13. Most failure cases stem from the inherent constraints of the generative model, particularly its suboptimal accuracy in multi-object generation, which occasionally leads to irrecoverable artifacts in the final output. Additionally, some errors arise from the VLM's judgment, where semantically subtle or ambiguous prompts are incorrectly validated, preventing VisualPrompter from identifying and optimizing these issues. Nevertheless, we anticipate that advancements in VLM capabilities will progressively mitigate these challenges, further enhancing the robustness of our model.

## H    INFLUENCE OF RANDOM SEED

To investigate the impact of random seeds, we conduct extensive experiments on two text-to-image generative models using five different random seeds and present the comprehensive results in Table 10. The experimental outcomes show remarkably consistent performance across all trials, with only minimal deviations observed. This stability clearly demonstrates that the influence of random seeds on our results is statistically negligible.

The observed robustness can be attributed to two key factors. First, the DSG benchmark incorporates an extensive set of over a thousand diverse prompts, providing sufficient data volume to effectively

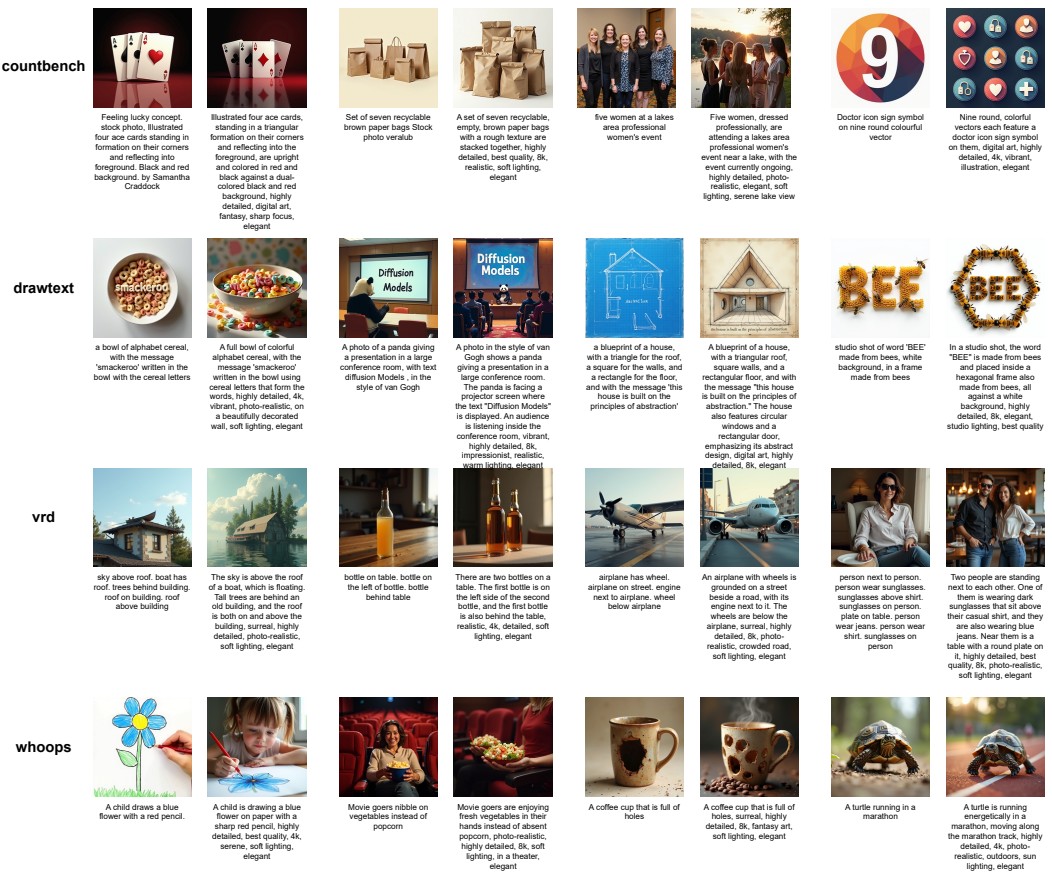

Figure 11: Examples of images generated by FLUX-dev. Each line corresponds to the results of prompts derived from distinct sub-datasets. The right image in each pair is generated from the prompt optimized by VisualPrompter, as detailed below the image.

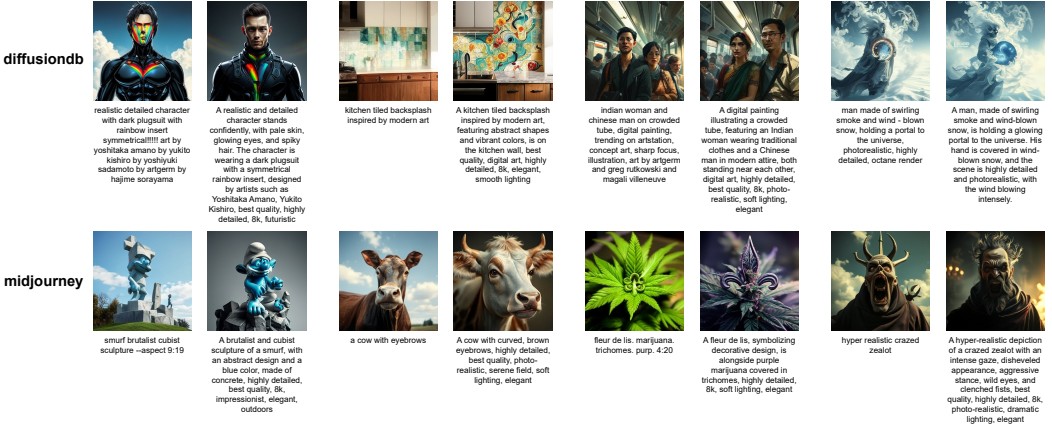

Figure 12: Examples of images generated by Janus-Pro. Each line corresponds to the results of prompts derived from distinct sub-datasets. The right image in each pair is generated from the prompt optimized by VisualPrompter, as detailed below the image.

average out any potential randomness. Second, while random variations might affect certain aspects like layout generation, they have minimal impact on semantic content representation. This is particularly relevant as our VisualPrompter framework is specifically designed to enhance and com-

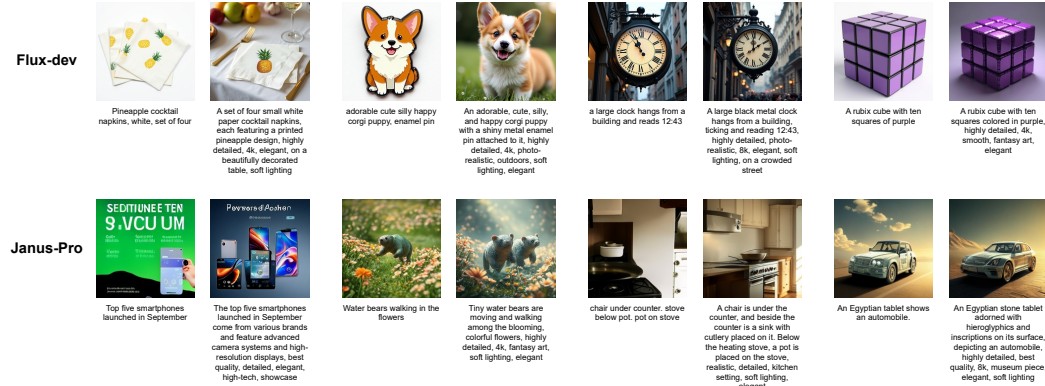

Figure 13: Some failure cases of VisualPrompter. The two rows of images are from Flux-dev and Janus-pro, respectively.

Table 10: Impact of random seed. We report the mean and standard deviation of semantic accuracy based on the DSG benchmark.

| T2I Models | Prompt Modification Approach | | |
| --- | --- | --- | --- |
| | None | Qwen3-32B | VisualPrompter |
| SD v1.5 | $67.4 \pm 0.13$ | $60.9 \pm 0.43$ | $69.8 \pm 0.56$ |
| FLUX-dev | $79.2 \pm 0.08$ | $79.7 \pm 0.21$ | $84.3 \pm 0.12$ |

plement semantic information processing, further mitigating any potential variability from random initialization.

## I    POTENTIAL TOWARD MULTIMODAL ANALYSIS

To evaluate the broader applicability of our approach, we integrate ControlNet (Zhang et al., 2023) with human pose estimation, enabling image generation conditioned on both skeletal inputs and textual prompts. We quantitatively assess skeleton-to-image consistency using our VisualPrompter's VLM module, which additionally provides refinement suggestions for improved alignment.

As illustrated in Figure 14, our framework demonstrates promising potential for multimodal analysis—effectively bridging visual structural data (skeletons) and linguistic descriptions. This multimodal capability suggests significant versatility in handling diverse input modalities. We identify this as a pivotal research direction and plan to systematically explore: (a) cross-modal interaction mechanisms, (b) scalability to additional modalities, and (c) quantitative evaluation frameworks, which will constitute a cornerstone of our future work.

## J    USER STUDY SETTINGS AND MORE COMPARISON

We provide a simple test system for our human evaluation experiment based on *streamlit*, as shown in Figure 15. Departing from the experiments in previous text-to-image prompt engineering studies that only focus on aesthetic evaluation, our assessment framework specifically incorporates an analysis of semantic alignment between generated visual content and the corresponding textual prompts, which measures how well the generated images align with the user intent.

As shown in Figure 15, a raw prompt and a pair of images are given to the participants for each test. The images pair consists of an image generated by the original prompt above and another image generated by the optimized prompt. In our experiment, the optimized prompts are invisible to the participants. To eliminate potential bias, images in each pair are randomly assigned to left or right positions. Participants are required to evaluate two key aspects individually: the visual consistency

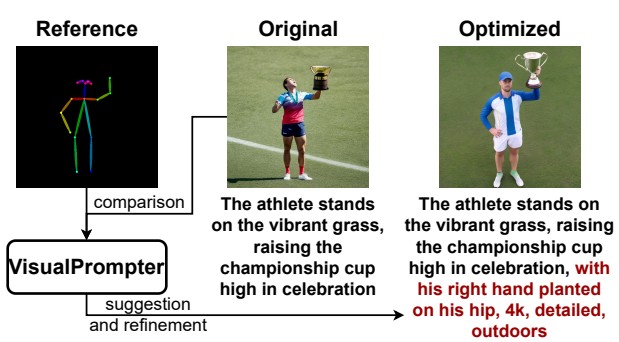

Figure 14: Multimodal application of VisualPrompter.

Figure 15: Screenshot of human evaluation interface.

Table 11: Comparison of VisualPrompter with other optimized methods. User preferences are measured as the percentage of choices for VisualPrompter, the competing model, or a tie, respectively.

| T2I Models | Aspects | Methods in Comparison | | |
| --- | --- | --- | --- | --- |
| | | NeuroPrompts | Promptist | BeautifulPrompt |
| SD v1.5 | Semantic | **58**% / 24% / 18% | **52**% / 27% / 21% | **60**% / 26% / 14% |
| | Aesthetics | 41% / **43**% / 16% | **44**% / 39% / 17% | 35% / **46**% / 19% |
| FLUX-dev | Semantic | **44**% / 30% / 26% | **43**% / 39% / 18% | **44**% / 31% / 25% |
| | Aesthetics | 34% / **46**% / 20% | 37% / **41**% / 22% | 27% / **49**% / 24% |

with the given prompt and the aesthetic appeal of each image. These evaluations provide essential insights into the overall quality of the visual representation.

To evaluate the semantic consistency and aesthetic quality of our model, we conduct the user study above based on Stable Diffusion v1.5 and Flux-dev, comparing it with other optimized models rather than the original generators. As illustrated in Table 11, the results demonstrate a clear user preference for our model in terms of semantic consistency. Furthermore, it achieves competitive performance on aesthetic metrics, underscoring its comprehensive capability.

## K    COMPARISON ON COMPLEX PROMPTS

To investigate performance on complex prompts, we collected additional prompts describing intricate scenes and generated images using Flux-dev, a generative model capable of handling complex scenarios. We then compared these results with outputs from our VisualPrompter-optimized prompts. As shown in Figure 16, our model demonstrates a clear ability to correct missing or erroneous semantics in the generated images. We also observed that the capability of the underlying generative model is crucial. Since long prompts already contain substantial detail, our optimization approach remains constrained when dealing with structures that are inherently challenging for the base generative model. We plan to continue research in this direction.

To validate the effectiveness of our method in optimizing multilingual text-to-image generation, we use Chinese text accepted by Qwen for testing. Since Stable Diffusion v1.5 and FLux-dev do not support Chinese, we select the text-to-image model Kolors (Kolors, 2024) as our subject and modify the prompt templates in VisualPrompter to support Chinese input. Visualization examples are shown in Figure 17. The results indicate that our method can also enhance Chinese text-to-image generation. This effectiveness stems from our language-agnostic atomic semantic structures and analytical approach, allowing for the processing of different languages supported by the underlying LLM.

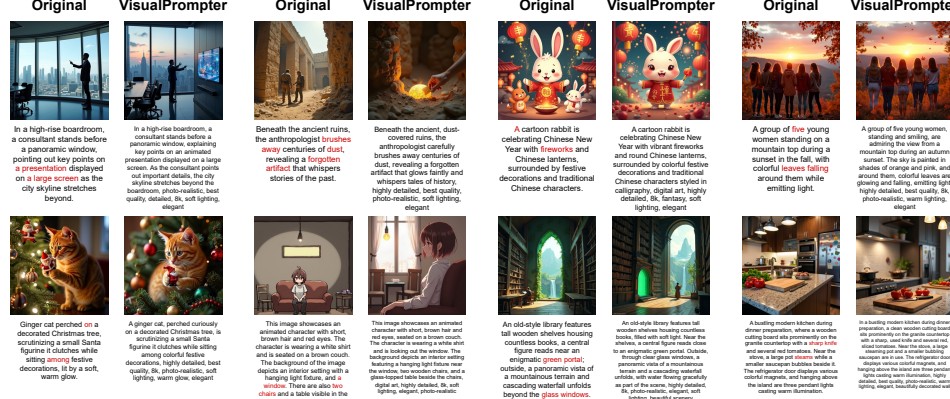

Figure 16: Visual comparison between the original Flux-dev (left) and our optimized framework (right) on complex prompts.

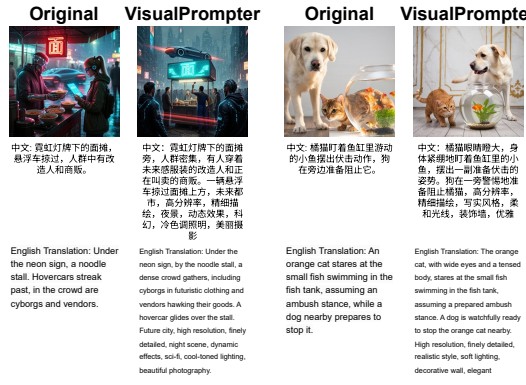

Figure 17: Examples of multilingual optimization based on Chinese, validating the effectiveness of the proposed method across languages. We employ Kolors (Kolors, 2024) as the image generator.

Table 12: Summarized results on the DSG benchmark using Llama-Vision. Reported scores are based on the percentage of "yes" answers to the questions, while the best scores are highlighted in boldface.

| Methods | DSG benchmark | | | | Average |
|---------|--------|--------|---------|-----------|---------|
| | SD 1.5 | SD 2.1 | Flux-dev | Janus-Pro | |
| Baseline | 67.2 | 72.0 | 79.4 | 79.5 | 74.5 |
| NeuroPrompts | 59.0 | 69.5 | 77.2 | 81.4 | 71.8 |
| Promptist | 65.9 | 68.9 | 77.6 | 78.2 | 72.7 |
| BeautifulPrompt | 49.4 | 51.1 | 53.4 | 57.0 | 52.7 |
| VisualPrompter | **70.1** | **77.9** | **85.4** | **82.9** | **79.1** |

## L ROBUSTNESS OF EVALUATION

We notice that the use of the same VLM for both the internal optimization feedback and the final evaluation may introduce potential bias. To address this concern and validate the robustness of our reported performance, we conduct an additional experiment where the results are re-evaluated using another VLM. We employ Llama-3.2-11B-Vision for a comparative analysis, as it is both open-source and readily accessible. As summarized in Table 12 and Table 13, the evaluation scores between the two VLMs are highly consistent. Despite minor fluctuations in evaluation scores, the

Table 13: Summarized results on the TIFA benchmark using Llama-Vision. Reported scores are based on the percentage of "yes" answers to the questions, while the best scores are highlighted in boldface.

| Methods | TIFA benchmark | | | | Average |
|---|---|---|---|---|---|
| | SD 1.5 | SD 2.1 | Flux-dev | Janus-Pro | |
| Baseline | 76.1 | 81.0 | 88.4 | 85.6 | 82.8 |
| NeuroPrompts | 64.3 | 77.2 | 85.5 | 89.8 | 79.2 |
| Promptist | 72.7 | 79.0 | 86.9 | 86.8 | 81.4 |
| BeautifulPrompt | 53.0 | 55.9 | 62.8 | 63.5 | 58.8 |
| VisualPrompter | **81.7** | **83.4** | **94.3** | **94.5** | **88.5** |

relative performance ranking of all optimization methods remains identical under both VLM assessors, underscoring that our comparative conclusions are robust to the choice of evaluator.

## M    DISCLOSURE OF LLM USE

The authors confirm that the use of LLMs, which was limited exclusively to polishing sentence structures and correcting grammatical errors without being involved in generating ideas, contributions, data, or any substantive content.

