# OpenReview forum: "VisualPrompter: Semantic-Aware Prompt Optimization with Visual Feedback for Text-to-Image Synthesis"
_ICLR.cc/2026/Conference — ICLR 2026 Poster_

### Official Review · Reviewer_5nxL · 2025-10-31

**Soundness:** 3
**Presentation:** 3
**Contribution:** 3
**Rating:** 6
**Confidence:** 4

**Summary:**

This paper addresses the semantic misalignment problem in text-to-image (T2I) generation, where sometimes model-preferred output images fail to match  user-provided prompts. The authors propose VisualPrompter, a training-free, semantic-aware prompt optimization framework that refines user prompts through atomic-level semantic decomposition based on the Davidsonian Scene Graph (DSG).
By parsing prompts into fine-grained semantic concepts through the Self-Reflection Module (SERE) and reconstructing coherent, model-preferred prompts via Target-Specific Prompt Optimization (TSPO), the method effectively produces prompts that better capture user intent and enhance text–image semantic alignment.The addressed problem is highly practically relevant for creative image generation and the proposed approach is easy to reproduce in real-world applications

**Strengths:**

- By explicitly addressing the problem of semantic omissions, the authors provide a fresh direction for prompt engineering research, shifting the focus from “visual beauty” to semantic faithfulness.By detecting and repairing semantic omissions between user text and generated images, the framework improves  intent alignment, which are crucial for real-world creative and design applications.
- The approach of decomposing prompts into atomic semantic units (entities, attributes, relations), using a Visual-Language Model (VLM) to detect absent ones, and then reassembling an optimized prompt with an LLM is conceptually new and technically creative  on the target problem. The integration of LLM reasoning and VLM verification within a self-reflective pipeline demonstrates strong originality.
- The use of two authoritative benchmarks, DSG-1k (ICLR 2024) and TIFA v1.0 (ICCV 2023), is appropriate and technically justified, as both measure fine-grained semantic alignment between text and image — directly matching the paper’s objective.

**Weaknesses:**

- Limited diversity of baselines: All three comparative methods (NeuroPrompts, Promptist, BeautifulPrompt)  share similar reinforcement-learning-based optimization paradigms.  The omission of  other omitted categories may weaken the empirical scope.
- Improvements over the baseline are modest (≈ 4–5 points on DSG/TIFA).  Given that VisualPrompter adds several modules and increases inference time (Table 6), the cost–benefit balance remains questionable.Since all reasoning and evaluation rely on Qwen2 and Qwen2-VL, it is also unclear how robust the approach is under different model backbones.
- The paper provides limited empirical evidence on the effectiveness of individual modules. More detailed analyses would strengthen the work — for example, examining which concept types (e.g., objects, attributes, or relations) or which modules (SERE vs TSPO) contribute most to performance gains.

**Questions:**

1. Please clarify the definition  of “Baseline.”
What exactly is the “Baseline” in Tables 1–3?  Is it simply the raw user prompt or an internal standardized prompt? Why does this baseline outperform some optimized methods?
2. Please justify the selection of comparison methods.
  Given that the Baseline achieves higher semantic alignment scores than other existing methods, please explain why NeuroPrompts, Promptist, and BeautifulPrompt were chosen as baselines of optimized methods.  Additionally, why were other families of methods (eg, multi-objective optimization methods, diffusion-specific) not considered for inclusion in the comparison?
3. Robustness and Ablation
  The paper would benefit from a more detailed ablation analysis showing the relative contribution of each module. Which component—SERE (Self-Reflection) or TSPO (Target-Specific Prompt Optimization) drives the largest improvements across different types of semantic concepts (e.g., objects, attributes, relations)? It would also be helpful to report whether the model’s gains are consistent across different diffusion backbones (e.g., SD v1.5 vs Flux-dev) and whether any module exhibits degradation or instability when applied to more complex or compositional prompts.

---

> ### Author Response · Authors · 2025-11-23
> **Response to Reviewer 5nxL (part 1/3)**
>
> **We are deeply grateful for the time and effort you have invested in reviewing our paper. Your valuable questions and insights have significantly contributed to enhancing our work. We respond to each comment as follows and sincerely hope that our rebuttal could properly address your concerns.**
>
> ## Weakness 1
> > Limited diversity of baselines: All three comparative methods (NeuroPrompts, Promptist, BeautifulPrompt) share similar reinforcement-learning-based optimization paradigms. The omission of other omitted categories may weaken the empirical scope.
>
> Thank you for raising this point. Our primary objective is to develop a training-free strategy to refine the user prompts for optimized image generation. Therefore, methods that involve fine-tuning generative models, as well as those that adjust object positions using additional modules or optimization, are excluded due to their inability to establish a fair comparison. In particular, our study focuses on state-of-the-art, publicly available methods that operate exclusively through prompt modification. In contrast to these studies that employ fine-tuned LLMs as prompt rewriters, our approach is totally training-free, which presents a significant difference.
> To mitigate the concern of limited baseline diversity, we incorporated an additional method TIPO [1] and presented its results in Table 14. Representative samples from TIPO can be found in Figure 16. We summarized the key comparisons in the tables below, where the values in parentheses indicate the performance difference of our VisualPrompter over TIPO.
>
> | On DSG bench   | SD v1.5      | SD v2.1      | Flux-dev     | Janus-Pro    | Average      |
> |----------------|--------------|--------------|--------------|--------------|--------------|
> | TIPO           | 54.4         | 62.1         | 70.1         | 70.3         | 64.2         |
> | VisualPrompter | 69.5 (+15.1) | 77.0 (+14.9) | 84.3 (+14.2) | 82.6 (+12.3) | 78.3 (+14.1) |
>
>
> | On TIFA bench  | SD v1.5      | SD v2.1      | Flux-dev     | Janus-Pro    | Average      |
> |----------------|--------------|--------------|--------------|--------------|--------------|
> | TIPO           | 61.3         | 66.6         | 79.5         | 79.1         | 71.6         |
> | VisualPrompter | 80.7 (+19.4) | 82.8 (+16.2) | 93.8 (+14.3) | 93.7 (+14.6) | 87.8 (+16.2) |
>
> The results demonstrate that our method outperforms TIPO by being more generalizable and semantically faithful, which substantiates its overall effectiveness. A detailed analysis is provided in our Appendix L.
>
>
> ## Weakness 2
> > Improvements over the baseline are modest (≈ 4–5 points on DSG/TIFA). Given that VisualPrompter adds several modules and increases inference time (Table 6), the cost–benefit balance remains questionable.Since all reasoning and evaluation rely on Qwen2 and Qwen2-VL, it is also unclear how robust the approach is under different model backbones.
>
> Thank you for your thoughtful comment regarding the cost-benefit balance and the robust of evaluation with VLM. We appreciate your valuable feedback and will provide a careful response to each of your points.
>
> **1. Modest Improvements.**
> We agree that this is a crucial consideration for any applied method. Our research deliberately focuses on a critical yet under-explored objective: optimizing for semantic consistency. As Figure 1 in our paper reveals, most existing works inadvertently compromise semantic fidelity in their pursuit of aesthetic enhancement. In this context, VisualPrompter's consistent and reliable improvement in semantic alignment (4-5 points on established benchmarks) is not merely a modest gain but a significant qualitative leap. It demonstrates that it is possible to bridge the gap between user intent and model output without sacrificing core content, thereby validating the importance and viability of this research direction.
>
> **2. Increased inference time.**
> Despite our workflow incorporates several additional steps, the bulk of the total generation time is dominated by image generation itself. In specific, when using FLUX as the image generator, all analysis and processing steps account for only 22.4% (7.7s) of the total 34.3s runtime. Another promising application of our work is in interactive workflows, where users can refine prompts directly using our VisualPrompter based on unsatisfactory images, eliminating the need for an extra generation cycle and thereby saving time.

---

> ### Author Response · Authors · 2025-11-23
> **Response to Reviewer 5nxL (part 2/3)**
>
> **3. Robust Under Different Backbones.**
> We select Qwen2-VL due to its state-of-the-art performance and high correlation with human judgment. To rigorously quantify any potential bias and directly address your concern about the robust under different model backbones, we conduct an additional experiment using a completely different open-source VLM, Llama-3.2-11B-Vision, to re-evaluate the generated results optimized by our VisualPrompter, which employs Qwen2-VL in its self reflection module. The results on the DSG benchmark are presented below and in Table 12 of our revision. Additionally, the results on the TIFA benchmark are summarized in Table 13. All values in the table report the Semantic Accuracy metric in percentage, with the parenthetical values indicating the deviation from the Qwen2-VL evaluation results, as shown in Table 1.
>
> | Methods           | SD 1.5            | SD 2.1            | Flux-dev          | Janus-Pro         | Average              |
> |-------------------|-------------------|-------------------|-------------------|-------------------|----------------------|
> | Baseline          |  67.2 (-0.3)      |  72.0 (-0.1)      |  79.4 (+0.3)      |  79.5 (+0.0)      |  74.53 (-0.025)      |
> | NeuroPrompts      |  59.0 (+0.6)      |  69.5 (+0.8)      |  77.2 (+1.7)      |  81.4 (-0.3)      |  71.75 (+0.700)      |
> | Promptist         | 65.9  (+0.8)      | 68.9  (+0.2)      | 77.6  (+0.7)      | 78.2  (+0.2)      | 72.65  (+0.475)      |
> | BeautifulPrompt   | 49.4  (+1.5)      | 51.1  (+1.5)      | 53.4  (+1.9)      | 57.0  (+1.7)      | 52.73  (+1.650)      |
> | **VisualPrompter**| **70.1**  (+0.6)  | **77.9**  (+0.9)  | **85.4**  (+1.1)  | **82.9**  (+0.3)  | **79.08**  (+0.967)  |
>
> As shown in the table above, the evaluation scores between the two VLMs are highly consistent. More importantly, VisualPrompter still achieves significant improvement under Llama's assessment, confirming that the impact of VLM-specific bias is minimal and that our conclusions remain robust across different evaluators. This experiment strengthens our claim that VisualPrompter provides a model-agnostic and reliable improvement in semantic alignment.
>
>
> ## Weakness 3
> > The paper provides limited empirical evidence on the effectiveness of individual modules. More detailed analyses would strengthen the work — for example, examining which concept types (e.g., objects, attributes, or relations) or which modules (SERE vs TSPO) contribute most to performance gains.
>
> We sincerely thank you for raising these important questions regarding module contributions and concept-level analysis. We have expanded our ablation study in Table 5 to dissect the contribution of each module. For your convenience, we have also included the table below (reporting Semantic Accuracy / Aesthetic Score).
>
> | SERE  | TSPO  | Result             |
> |-------|-------|--------------------|
> | w/o   | w/o   | 72.1 / 5.48        |
> | w/    | w/o   | 73.8 / 5.54        |
> | w/o   | w/    | 71.9 / 5.69        |
> | w/    | w/    | **77.0** / **5.73**|
>
> The results clearly demonstrate the crucial synergy between the two components. The SERE module serves as a diagnostic tool, effectively identifying the specific concepts missing in the generated images, which is the essential first step for targeted improvement. Although the TSPO module alone does not yield significant improvements, its combination with SERE substantially enhances overall performance. The close collaboration between the two modules is key to VisualPrompter’s success.
> While our current findings robustly establish their synergistic roles, we agree that a fine-grained analysis by concept type would be informative. We will assess their distinct impacts in future work to complete our understanding.
>
> ## Question 1
> > Please clarify the definition of “Baseline.” What exactly is the “Baseline” in Tables 1–3? Is it simply the raw user prompt or an internal standardized prompt? Why does this baseline outperform some optimized methods?
>
> We sincerely thank you for the constructive feedback. We will address all of your points systematically in the responses below.
>
> **1. What exactly is the baseline in Tables 1–3?**
> In Tables 1–3, the "baseline" refers to the raw user prompts without any optimization—that is, the original textual descriptions provided by users, which are directly fed into the generative models for image synthesis. This baseline is not an internal standardized prompt, but the unmodified input from benchmark datasets (DSG and TIFA).

---

> ### Author Response · Authors · 2025-11-23
> **Response to Reviewer 5nxL (part 3/3)**
>
> **2. Why does this baseline outperform some optimized methods?**
> A key finding of our work is a significant gap in current T2I prompt optimization research: while existing methods effectively enhance aesthetic quality (as shown in Table 3, where all compared methods outperform the baseline in aesthetic evaluation), they often do so at the cost of semantic consistency. This results in visually pleasing images that nonetheless fail to align with the user's original intent. This trade-off is clearly reflected in our fine-grained semantic evaluation, where these methods exhibit a noticeable drop in semantic accuracy.
> As analyzed in Section 4.3, methods such as NeuroPrompts and Promptist tend to introduce irrelevant keywords, while BeautifulPrompt frequently omits critical information from the original prompt. In contrast, our VisualPrompter is explicitly designed to preserve original semantic content while strategically enhancing missing concepts. This dual focus enables it to consistently surpass both the baseline and existing methods in semantic accuracy, without compromising aesthetic improvement.
>
>
> ## Question 2
> > Please justify the selection of comparison methods. Given that the Baseline achieves higher semantic alignment scores than other existing methods, please explain why NeuroPrompts, Promptist, and BeautifulPrompt were chosen as baselines of optimized methods. Additionally, why were other families of methods (eg, multi-objective optimization methods, diffusion-specific) not considered for inclusion in the comparison?
>
> Thank you very much for your thoughtful comment. As detailed in Weakness 1 above, the methods we have chosen for comparison—NeuroPrompts, Promptist, and BeautifulPrompt—represent state-of-the-art, publicly available approaches that are strictly focused on prompt rewriting for text-to-image generation, which allows for a direct and fair evaluation within the scope of prompt engineering. Other families of methods, such as multi-objective optimization or diffusion-specific techniques, often involve model fine-tuning and reinforcement learning, making them less suitable for a controlled comparison focused purely on prompt optimization.
>
> ## Question 3
> > Robustness and Ablation The paper would benefit from a more detailed ablation analysis showing the relative contribution of each module. Which component—SERE (Self-Reflection) or TSPO (Target-Specific Prompt Optimization) drives the largest improvements across different types of semantic concepts (e.g., objects, attributes, relations)? It would also be helpful to report whether the model’s gains are consistent across different diffusion backbones (e.g., SD v1.5 vs Flux-dev) and whether any module exhibits degradation or instability when applied to more complex or compositional prompts.
>
> We sincerely thank you for raising these important suggestions. We will address all of your questions and comments in the responses below.
>
> **1. Detailed ablation analysis of each module.**
> As detailed in response to Weakness 3, our expanded ablation studies confirm the complementary roles of SERE and TSPO: SERE is key to diagnosing the missing concepts, while TSPO is crucial for addressing them, and their combined operation yields significant gains.
>
> **2. Whether the model’s gains are consistent across different diffusion backbones.**
> Detailed results on the DSG and TIFA benchmarks, as shown in Tables 8 and Table 9, demonstrate that our method achieves consistent improvements across different generative models, including Stable Diffusion v1.5 and Flux-dev. However, the extent of improvement varies, which can be attributed to the inherent capabilities of each generative model.
>
> **3. Whether any module exhibits degradation or instability when applied to more complex or compositional prompts.**
> The addition of samples based on complex and compositional prompts shown in Figure 17 highlights our model's advanced prompt refinement capability. Our method utilizes the DSG structure, which provides a detailed and reliable atomic semantic representation that performs well on complex sentences. However, the effectiveness of our TSPO module can be compromised by long sentences, hindering its ability to construct a high-quality prompt. The failure cases in Figure 14 further underscore that the model's own generative capability is a crucial limiting factor; optimizations like ours can only offer limited gains if the base model struggles with lengthy and complex prompts.
>
> [1] Shih-Ying Yeh, Sang-Hyun Park, Giyeong Oh, Min Song, and Youngjae Yu. TIPO: text to image with text presampling for prompt optimization, 2024.

---

> ### Author Response · Authors · 2025-11-23
> **Brief Summary of Responses to Reviewer 5nxL**
>
> Dear Reviewer 5nxL,
>
> We are deeply grateful for the time and effort you have invested in reviewing our paper. To conserve your valuable time, here we summary the key points of our rebuttal to facilitate a quick read.
>
> - **Baseline Selection Justification:** We selected SOTA LLM-based prompt optimization methods for a direct and fair comparison within the specific task of prompt engineering for T2I generation. Furthermore, TIPO was included as an additional baseline to broaden the variety of compared methods.  (Please refer to **Weakness 1** in our response for details)
>
> - **Cost-Benefit and Robustness Validation:** The increased inference time is largely dominated by image generation itself, and our pipeline's analysis stages account for a minor portion (22.4% with Flux-dev). Additional evaluation using a different VLM (Llama-3.2-Vision) confirms our results are robust and not biased. (Please refer to **Weakness 2** in our response for details)
>
> - **Module Contribution Analysis:** Expanded ablation studies in Table 5 demonstrate that both the SERE and TSPO modules are crucial and complementary. (Please refer to **Weakness 3** in our response for details)
>
> - **Baseline Definition and Dropped Performance:** The baseline in our tables is the results of raw, unmodified user prompt from the benchmarks. The decline in performance indicates a key trade-off that existing methods often boost aesthetics at the cost of semantic fidelity, a gap our method specifically addresses. (Please refer to **Question 1** in our response for details)
>
> - **Selection of Comparison Methods:** Explained in the response to Weakness 1. (Please refer to **Question 2** in our response for details)
>
> - **Consistency Across Models and Complex Results:** Our method achieves consistent improvements across different diffusion backbones (SD v1.5, Flux-dev), though the degree of improvement varies with the base model's inherent capabilities. We have added new challenging and realistic examples in Figure 17 of our revised manuscript to better illustrate VisualPrompter's strong performance. (Please refer to **Question 3** in our response for details)
>
> We sincerely hope that our responses have adequately addressed the points you raised. If you have further concerns, please let us know and we will continue actively responding to your comments and improving our submission. We would deeply appreciate it if you could raise your score.
>
> Thank you very much for your invaluable time and thoughtful consideration.
>
> Best regards,
>
> Authors

---

### Official Review · Reviewer_wv9m · 2025-10-31

**Soundness:** 3
**Presentation:** 4
**Contribution:** 3
**Rating:** 4
**Confidence:** 3

**Summary:**

This paper introduces VisualPrompter, a training-free, plug-and-play framework that uses self-reflection to detect missing concepts and performs atomic-level prompt edits to improve semantic alignment between user descriptions and text-to-image outputs. Experiments show state-of-the-art results on multiple text–image alignment benchmarks.

**Strengths:**

1. Visual Prompter leverage visual-language models (VLMs) for question–answer-based detection of missing semantic concepts in generated images, aligns with human intuition and exhibits high interpretability.
2. Visual Prompter significantly outperforms current state-of-the-art prompt engineering methods in multiple benchmarks.

**Weaknesses:**

1. The user study compares Visual Prompter only with the baseline (original prompts), rather than with other prompt optimization methods.
2. Lacks comparison with recent methods, such as 《TIPO: Text to Image with Text Presampling for Optimal Prompting.》
3. In Figure 11, the original prompts themselves are ambiguous and unnatural for human expression, such as “person next to person” or “bottle on the left of bottle.” I would like to see the performance of VisualPrompter on more natural and human-like prompts, such as those mentioned in Figure 1.

**Questions:**

Questions：
1. VisualPrompter uses Qwen2-VL as the visual question answering model; however, SEMANTIC ACCURACY is also evaluated using Qwen2-VL as the assessment model. Would this introduce a bias?

---

> ### Author Response · Authors · 2025-11-23
> **Response to Reviewer wv9m (part 1/2)**
>
> **We thank you for the insightful and valuable comments and constructive suggestions. We have performed additional experiments and revised the manuscript accordingly. Here we respond to each comment as follows and sincerely hope that our rebuttal could properly address your concerns.**
>
> ## Weakness 1
> > The user study compares Visual Prompter only with the baseline (original prompts), rather than with other prompt optimization methods.
>
> Thank you for raising this important point. We appreciate your advice and supplement our experiments with comparisons against the other optimized methods on Stable Diffusion V1.5 and Flux-dev. We follow the same user study protocol described in our Appendix J. The results are shown below and in Table 11 of our newest revision. Each set of three percentages represents user preference: for our results, the baseline results, or for tie, respectively.
>
> | T2I Models   | Aspects   | Vs NeuroPrompts (ours / rival / tie) | Vs Promptist (ours / rival / tie) | Vs BeautifulPrompt(ours / rival / tie) |
> |--------------|-----------|-----------------------------------|--------------------------------|-------------------------------------|
> | **SD v1.5**  | Semantic  | **58 %** / 24 % / 18 %            | **52 %** / 27 % / 21 %         | **60 %** / 26 % / 14 %              |
> | **SD v1.5**  | Aesthetics| 41 % / **43 %** / 16 %            | **44 %** / 39 % / 17 %         | 35 % / **46 %** / 19 %              |
> | **FLUX-dev** | Semantic  | **44 %** / 30 % / 26 %            | **43 %** / 39 % / 18 %         | **44 %** / 31 % / 25 %              |
> | **FLUX-dev** | Aesthetics| 34 % / **46 %** / 20 %            | 37 % / **41 %** / 22 %         | 27 % / **49 %** / 24 %              |
>
> Our study reveals that while existing prompt optimization methods can indeed enhance the aesthetic quality of generated images, they often cause significant degradation in semantic alignment, resulting in visually polished images that nonetheless fail to match the user’s original intent. The results above confirm that our model achieves substantially better semantic consistency than these existing approaches, while also maintaining comparative aesthetics.
>
> ## Weakness 2
> > Lacks comparison with recent methods, such as 《TIPO: Text to Image with Text Presampling for Optimal Prompting.》
>
> We thank you for pointing out the recent TIPO method. In response, we conduct experiments using TIPO's publicly available weights (TIPO-200M-FT2-FP16) on the DSG benchmark and the TIFA benchmark. The summary is presented in the tables below, with detailed results provided in Table 14 of our revision. The values in parentheses are the performance differences of our VisualPrompter over TIPO.
>
> | On DSG bench   | SD v1.5      | SD v2.1      | Flux-dev     | Janus-Pro    | Average      |
> |----------------|--------------|--------------|--------------|--------------|--------------|
> | TIPO           | 54.4         | 62.1         | 70.1         | 70.3         | 64.2         |
> | VisualPrompter | 69.5 (+15.1) | 77.0 (+14.9) | 84.3 (+14.2) | 82.6 (+12.3) | 78.3 (+14.1) |
>
>
> | On TIFA bench  | SD v1.5      | SD v2.1      | Flux-dev     | Janus-Pro    | Average      |
> |----------------|--------------|--------------|--------------|--------------|--------------|
> | TIPO           | 61.3         | 66.6         | 79.5         | 79.1         | 71.6         |
> | VisualPrompter | 80.7 (+19.4) | 82.8 (+16.2) | 93.8 (+14.3) | 93.7 (+14.6) | 87.8 (+16.2) |
>
> The results demonstrate that our method still outperforms TIPO, which appears to struggle with generalizing effectively across diverse image generation tasks. Our deeper investigation indicates that TIPO's training data leads to specialized optimization for human figures and scenes, introducing numerous person-related keywords. We also provide several random cases of TIPO in Figure 16.
> These results demonstrate that the effectiveness of TIPO is not general, but is confined to specific domains such as animated characters and landscapes. This fact explains the significant performance gap between TIPO and our method. It also highlights a key strength of our training-free, feedback-driven approach, whose model-agnostic design does not inherit such distributional biases, enabling more generalizable and semantically faithful prompt optimization.

---

> ### Author Response · Authors · 2025-11-23
> **Response to Reviewer wv9m (part 2/2)**
>
> ## Weakness 3
> > In Figure 11, the original prompts themselves are ambiguous and unnatural for human expression, such as “person next to person” or “bottle on the left of bottle.” I would like to see the performance of VisualPrompter on more natural and human-like prompts, such as those mentioned in Figure 1.
>
> We appreciate your valuable comment. To better illustrate performance on natural, human-like prompts, we add several challenging and realistic examples in Figure 17 of our revised manuscript. These new results clearly demonstrate VisualPrompter's ability to effectively expand and optimize natural language descriptions. Additionally, the DSG benchmark includes results from natural prompts, as shown in the last line of Figure 11 and in Figure 12.
>
> ## Question 1
> > VisualPrompter uses Qwen2-VL as the visual question answering model; however, SEMANTIC ACCURACY is also evaluated using Qwen2-VL as the assessment model. Would this introduce a bias?
>
> Thank you for raising this important point regarding the potential bias of using Qwen2-VL in both the optimization and evaluation stages. To rigorously quantify any potential bias, we conduct an additional experiment using a completely different open-source VLM, i.e., Llama-3.2-11B-Vision, to re-evaluate the generated results optimized by our VisualPrompter, which employs Qwen2-VL in its self reflection module. The results on the DSG benchmark are presented below and in Table 12 of our revision. Additionally, the results on the TIFA benchmark are reported in Table 13. All values in the table report the Semantic Accuracy metric in percentage, with the parenthetical values indicating the deviation from the Qwen2-VL evaluation results, as shown in Table 1.
>
> | Methods           | SD 1.5            | SD 2.1            | Flux-dev          | Janus-Pro         | Average              |
> |-------------------|-------------------|-------------------|-------------------|-------------------|----------------------|
> | Baseline          |  67.2 (-0.3)      |  72.0 (-0.1)      |  79.4 (+0.3)      |  79.5 (+0.0)      |  74.53 (-0.025)      |
> | NeuroPrompts      |  59.0 (+0.6)      |  69.5 (+0.8)      |  77.2 (+1.7)      |  81.4 (-0.3)      |  71.75 (+0.700)      |
> | Promptist         | 65.9  (+0.8)      | 68.9  (+0.2)      | 77.6  (+0.7)      | 78.2  (+0.2)      | 72.65  (+0.475)      |
> | BeautifulPrompt   | 49.4  (+1.5)      | 51.1  (+1.5)      | 53.4  (+1.9)      | 57.0  (+1.7)      | 52.73  (+1.650)      |
> | **VisualPrompter**| **70.1**  (+0.6)  | **77.9**  (+0.9)  | **85.4**  (+1.1)  | **82.9**  (+0.3)  | **79.08**  (+0.967)  |
>
> As shown in the table above, the evaluation scores between the two VLMs are highly consistent. More importantly, VisualPrompter still achieves significant improvement under Llama's assessment, demonstrating that the impact of VLM-specific bias is minimal and that our conclusions remain robust across different evaluators.

---

> ### Author Response · Authors · 2025-11-23
> **Brief Summary of Responses to Reviewer wv9m**
>
> Dear Reviewer wv9m,
>
> We are deeply grateful for the time and effort you have invested in reviewing our paper. To conserve your valuable time, here we summary the key points of our rebuttal to facilitate a quick read.
>
> - **Additional Comparative User Study:** We have conducted additional user studies comparing our VisualPrompter against other methods. The results confirm that our method achieves better semantic consistency, while maintaining competitive aesthetic quality. (Please refer to **Weakness 1** in our response for details)
>
> - **Comparison with Recent Method (TIPO):** We have performed new comparative experiments with TIPO. The results demonstrate that our method significantly outperforms TIPO. (Please refer to **Weakness 2** in our response for details)
>
> - **Performance on Natural and Human-like Prompts:** We have added new challenging and realistic examples in Figure 17 of our revised manuscript to better illustrate VisualPrompter's strong performance. (Please refer to **Weakness 3** in our response for details)
>
> - **Evaluation Model Bias Analysis:** We have conducted additional experiments using a different VLM (Llama-3.2-Vision) to re-evaluate our results, confirming that the potential bias from using Qwen2-VL is minimal. (Please refer to **Question 1** in our response for details)
>
> We truly hope our responses have adequately addressed your concerns. We would deeply appreciate it if you could raise your score. If you have further concerns, please let us know and we will continue actively responding to your comments and improving our submission.
>
> Thank you for your invaluable time and thoughtful review.
>
> Best regards,
>
> Authors

---

### Official Review · Reviewer_cJ9v · 2025-11-01

**Soundness:** 3
**Presentation:** 3
**Contribution:** 2
**Rating:** 4
**Confidence:** 4

**Summary:**

The paper aims to improve the semantic alignment between generated images and user descriptions.


It proposes VisualPrompter, a training-free prompt engineering framework that iteratively refines user prompts. VisualPrompter has a self-reflection module that analyzes generated images, and a target-specific prompt optimization that revises the prompt later. The method is plug-and-play and achieves state-of-the-art alignment on multiple image generation benchmarks.

**Strengths:**

- Easy to use: the proposed VisualPrompter is model-agnostic and plug-and-play, making it highly adaptable to various generative models.
- Good results: VisualPrompter outperforms many baselines on multiple benchmarks and multiple generative models, as shown in Table 1.

**Weaknesses:**

- Auxiliary LLM bias: Introducing an additional LLM in the loop may inject its own biases, especially there’re multiple LLM calls.
- Compute overhead and latency: the generate - analyze - revise cycles may be significantly more expensive than a single forward pass. In addition, LLMs were called multiple times in one image generation, which might be costly.
 - Limited contribution: modules are not novel. For example, regarding the reflection module, the LLM Expander, the LLM Composer, similar techniques can be found in references. It'll be great if author could explain what's the unique contribution and novelty in VisualPrompter.

**Questions:**

- Robustness: How is the performance with difficult prompts, for example, long prompts, multilingual prompts, code-mixed prompts? Could you show any qualitative failure cases?

---

> ### Author Response · Authors · 2025-11-23
> **Response to Reviewer  cJ9v (part 1/2)**
>
> **We thank you for the insightful and valuable comments. We respond to each comment as follows and sincerely hope that our rebuttal could properly address your concerns.**
>
> ## Weakness 1
> > Auxiliary LLM bias: Introducing an additional LLM in the loop may inject its own biases, especially there're multiple LLM calls.
>
> We sincerely thank you for the valuable comments and constructive feedback. We have noticed the potential bias introduced by multiple LLMs and have employed the following three strategies to mitigate it:
> **1. Semantic Control via DSG Structure.**
> In VisualPrompter, all semantic content of the original prompt is governed by the Davidsonian Scene Graph (DSG) [1] structure. During both DSG generation and prompt reconstruction, all atomic concepts from the original user input are preserved, thus prevents the LLM from altering the user’s intent, reducing errors and improving the accuracy of output.
> **2. Targeted and Limited Concept Expansion.**
> Concept expansion is applied only to missing concepts identified by the visual feedback module. Each LLM is assigned a specific, localized task and does not interfere with other parts of the sentence or outputs from other LLMs in the pipeline, avoiding unnecessary modification of original user prompts.
> **3. Guided Generation with Templates and Examples.**
> We use carefully designed prompt templates accompanied by a substantial set of in-context examples to steer the LLM toward strictly adhering to the semantic structure of the original input. Examples of expanding semantic and composing prompt with LLM are provided in Figure 8 and Figure 9 of our Appendix.
>
> Quantitative and qualitative experimental results in Section 4.3 confirm that VisualPrompter effectively preserves user semantics while generating diverse and contextually enriched expansions, leading to the creation of high-quality and semantically faithful images.
>
>
> ## Weakness 2
> > Compute overhead and latency: the generate - analyze - revise cycles may be significantly more expensive than a single forward pass. In addition, LLMs were called multiple times in one image generation, which might be costly.
>
> We sincerely thank you for raising this critical point. As reported in Table 6, we have specifically measured this trade-off. Despite our workflow incorporates several additional steps, the bulk of the total generation time is dominated by image generation itself. In specific, when using Flux-dev as the image generator, all analysis and processing steps account for only 22.4% (7.7s) of the total 34.3s runtime.
>
> Notably, our VisualPrompter offers a practical optimization: when a user generates an image and is unsatisfied with the result, he can use our model to directly refine the prompt based on the unsatisfactory image and regenerate it, saving the time of an additional image generation cycle.

---

> ### Author Response · Authors · 2025-11-23
> **Response to Reviewer  cJ9v (part 2/2)**
>
> ## Weakness 3
> > Limited contribution: modules are not novel. For example, regarding the reflection module, the LLM Expander, the LLM Composer, similar techniques can be found in references. It'll be great if author could explain what's the unique contribution and novelty in VisualPrompter.
>
> Thank you for your valuable question. We provide a detailed description of the self-reflection module, Expander, and Composer modules, covering their roles in our framework and distinguishing them from existing approaches to address your concern.
>
> **1. Self Reflection Module (SERE) as Semantic Decomposer.**
> Previous works like DSG [1] utilizes self-reflection to evaluate the semantic consistency between generated images and given prompts, serving as a metric. However, we use it to derive feedback from visual content and incorporate it into the prompt optimization task to achieve enhanced performance.
>
> **2. LLM Expander as Context Enricher.**
> Prior LLM Expanders [2] typically operate at the sentence level, i.e., using a fine-tuned LLM to rewrite or expand the entire sentence. In contrast, our method operates at atomic semantic level. In particular, we guide the LLM to incorporate necessary new atomic semantics that are relevant to specific targets, leaving other existing context unchanged. This enables adding reasonable new semantic information, preserving the original user intent as much as possible and offering better interpretability for detailed modifications.
>
> **3. LLM Composer as Prompt Rewritter.**
> Existing LLM composers [3] typically focus on object-centric semantic relationships to construct a complete scene (depicting spatial relations and overall structure), while our composer targets at faithfully reconstructing a complete sentence from its atomic semantic components. Thanks to its well-defined semantic structure and dependencies, the generated sentences exhibit high readability and are preferred by generation models as demonstrated in Table 1 and Figure 3.
>
> In conclusion, our VisualPrompter pioneers a novel direction in prompt optimization by refining user prompts through atomic semantic expansion. The core novelty of our work lies in its unique leverage of visual feedback to identify shortcomings in generated images, coupled with a breakdown-and-compose paradigm for semantic enrichment. The integration of these components into a novel end-to-end pipeline constitutes our primary innovation.
>
> ## Question 1
> > Robustness: How is the performance with difficult prompts, for example, long prompts, multilingual prompts, code-mixed prompts? Could you show any qualitative failure cases?
>
> Thank you for this feedback. To further address your question about challenging prompts, we have included additional examples involving long and complex prompts in Figure 17 of the updated version. To further demonstrate the multilingual capability of our method, we include optimization results based on Chinese prompts in Figure 18. A more detailed analysis is provided in Appendix M.
> Failure cases have been shown in Figure 13, which reveal two primary types of errors. The first type originates from the fundamental constraints of the generative models themselves. The second type occurs when our semantic expansion introduces descriptions that are inconsistent with the visual context, thereby failing to rectify the original issue. Our quantitative statistics shows that Type I errors constitute the majority (>80%) of the failures. Addressing both error types is a key objective of our future work, which will focus on enhancing the factual grounding of our expansions and adopting more powerful generative models.
>
> [1] Jaemin Cho, Yushi Hu, Jason M. Baldridge, Roopal Garg, Peter Anderson, Ranjay Krishna, Mohit Bansal, Jordi Pont-Tuset, and Su Wang. Davidsonian scene graph: Improving reliability in fine-grained evaluation for text-to-image generation. ICLR 2024.
> [2] Yaru Hao, Zewen Chi, Li Dong, and Furu Wei. Optimizing prompts for text-to-image generation. NeurIPS 2023.
> [3] TsungHan Wu, Long Lian, Joseph E. Gonzalez, Boyi Li, and Trevor Darrell. Self-correcting llm-controlled diffusion models. CVPR 2024.

---

> ### Author Response · Authors · 2025-11-23
> **Brief Summary of Responses to Reviewer cJ9v**
>
> Dear Reviewer cJ9v,
>
> We are deeply grateful for the time and effort you have invested in reviewing our paper. To conserve your valuable time, here we summary the key points of our rebuttal to facilitate a quick read.
>
> - **Auxiliary LLM Bias Mitigation:** We address potential bias from multiple LLM calls through a structured DSG framework, targeted concept expansion, and guided generation with templates, ensuring semantic fidelity and reducing errors. (Please refer to **Weakness 1** in our response for details)
>
> - **Compute Overhead and Latency Management:** Our experiments show the analysis and processing by LLMs and VLM constitutes only 22.4% of the total runtime when using Flux-dev as generator, while the majority of the time is spent on the image generation process. (Please refer to **Weakness 2** in our response for details)
>
> - **Clarification on Novelty and Contribution:** The core novelty lies in our unique paradigm of using visual feedback for atomic-level semantic expansion and recomposition, moving beyond sentence-level rewriting. (Please refer to **Weakness 3** in our response for details)
>
> - **Robustness and Failure Cases:** We have added qualitative results for complex and multilingual prompts. Failure cases have been provided and analyzed in the revision. (Please refer to **Question 1** in our response for details)
>
> We humbly hope that our responses have adequately addressed your concerns. We would deeply appreciate it if you could raise your score. If you have further concerns, please let us know and we will continue actively responding to your comments and improving our submission.
>
> Thank you for your invaluable time and consideration.
>
> Best regards,
>
> Authors

---

### Author Response · Authors · 2025-12-03
**Rebuttal Summary for Paper 19539**

Dear Area Chair,

In light of the suspended discussion, we sincerely appreciate your diligence and thoughtful effort in evaluating our submission. To support your meta-review, we provide below a faithful and brief summary of the rebuttal process. Thank you very much for your time and attention.

We also extend our gratitude to the three reviewers (cJ9v, wv9m, and 5nxL) for their valuable and constructive feedback. All reviewers recognized the core strengths of VisualPrompter, praising **our fresh research direction** (5nxL), **technically creative method** (wv9m and 5nxL), **strong performance** (cJ9v and wv9m), **easy-to-use framework** (cJ9v), and **technically justified evaluation methodology** (5nxL).

**We have done our utmost to address all reviewer concerns and no further issues were raised up to the point of termination.**

First, regarding our methodological rationale (raised by cJ9v and 5nxL), our core contribution lies in **proposing a novel and training-free atomic-level semantic expansion paradigm, distinct from existing sentence-level rewriting approaches**. Ablation study (Table 5) proved the synergistic importance of both our core modules.

Second, for the concern about the efficiency-performance trade-off (noted by cJ9v and 5nxL), our experiments show that VisualPrompter introduces only minor computational overhead (e.g., **22.4%** of total runtime with Flux-dev) while **achieving a significant gain on the crucial metric of semantic accuracy**. Further experiments (Tables 2, 8 and 9) demonstrated the improvement remains consistent across different diffusion backbones.

Finally, to address all the reviewers' requests for comprehensive and robust experimental validation, we have conducted a series of additional experiments. These include additional results on long and complex cases (Figure 17) and multilingual cases (Figure 18); a further user study confirming our method's advantage in semantic accuracy (Table 11, with a **12.3%** win rate); direct comparisons with methods like TIPO (Table 14, VisualPrompter outperforms it by over **14.1%** on DSG and **16.1%** on TIFA benchmarks); and confirming the robustness of both the evaluation metrics and our performance gains by using a different VLM for re-evaluation (Tables 12 and 13).

We are confident that the comprehensive revisions and supplementary analyses have significantly strengthened the rigor and empirical support of our work. The revised manuscript now fully addresses the reviewers' concerns and would merit a positive recommendation. We are grateful that the constructive review process has greatly improved our work.

Thank you again for your efforts in handling our submission.

Sincerely,
Authors of Paper 19539

---

### Meta-Review · Area_Chair_NkcM · 2026-01-05

**Summary:**

The initial reviews were mixed but lean very slightly towards negative (6 / 4 / 4). On the positive side, the reviewers praised the approach (using a VLM for VQA on the resulting image and prompt to iteratively improve the prompt) as well as the results. For concerns, reviewers raised concerns about the novelty of the approach, the thoroughness of the baselines included, and the amount of extra inference time being added.

**Reviewer Concerns:**

- On the novelty of the approach, the authors indicate that the novelty lies specifically in the design of their modules: Self Reflection module as Semantic Decomposer, LLM Expander as Context Enricher, and LLM Composer as Prompt Rewritter as well as the core novelty being "the core novelty of our work lies in its unique leverage of visual feedback to identify shortcomings in generated images, coupled with a breakdown-and-compose paradigm for semantic enrichment. The integration of these components into a novel end-to-end pipeline constitutes our primary innovation".
- On the thoroughness of the baselines, the authors give a two pointed answer, 1) saying that their explicit goal was to create a training-free strategy which rule out many of the potential baseline comparisons (such as multi-objective optimization or diffusion-specific techniques), 2) adding a comparison to TIPO in one table and NeuroPrompts, Promptist, and BeautifulPrompt in another. The table for NeuroPrompts, Promptist, and BeautifulPrompt shows gains for Semantics, occasionally at the cost of Aesthetics.
- On the amount of time added, the time added is around 22.4% percent with the T2I model dominating that amount of time.

**Reviewer Scores:**

Generally, I believe that the novelty and amount of time added are sufficiently addressed by the author's rebuttal. I think the baselines response was sufficient for the papers the reviewers mentioned though not fully compelling. My general belief is that both 4 may have raised to 6.

---

### Decision · Program_Chairs · 2026-01-26

Accept (Poster)